# Development and Characterization of Clay–Nanocomposites for Water Purification

**DOI:** 10.3390/ma13173793

**Published:** 2020-08-27

**Authors:** Richard Barnes Asamoah, Abu Yaya, Paschal Nbelayim, Ebenezer Annan, Boateng Onwona-Agyeman

**Affiliations:** Department of Materials Science & Engineering, School of Engineering Sciences, CBAS, University of Ghana, Legon, Accra 00233, Ghana; rbasamoah001@st.ug.edu.gh (R.B.A.); ayaya@ug.edu.gh (A.Y.); sugril2002@yahoo.com (P.N.); ebannan@ug.edu.gh (E.A.)

**Keywords:** clay–nanocomposites, antibacteria activity, water purification

## Abstract

In this work we propose a facile method of preparing cost-effective clay-metaloxide/metal (CuO/Ag and ZnO/Ag) composite pellets for an efficient water purification technique. Clay, with intrinsic antibacterial activity, served as a membrane support for different metaloxide/Ag nanoparticles (NPs) concentrations (2.5, 5 and 10 wt.%), as the active fillers. The effect of time (24 and 48 h) on the bactericidal activity of these pellets was also monitored. The clay–nanocomposite pellets were characterized using: X-ray diffraction, X-ray fluorescence, scanning electron microscopy, transmission electron microscopy, ultraviolet–visible (UV–Vis) spectrophotometry and nitrogen desorption analysis. The antibacterial activity performance was tested using *E. coli* and *S. aureus* strains of ATCC25922 and ATCC25923, respectively, in two aqueous media (nutrient broth and nutrient-free) by the colony-forming unit method. The results showed that the clay-CuO/Ag composite with a bandgap (1.24 eV) exhibited overall best performance under all conditions and time factors of ~100% efficiency in nutrient-free medium for all concentrations and times and 20–40% efficiency in nutrient broth for 24 h. The clay-ZnO/Ag with a bandgap of 2.88 eV showed no bactericidal activity in both media, except for that with 10 wt.% ZnO/Ag which showed 100% efficiency in nutrient-free medium after 24 h. All the synthesized composites showed 100% bactericidal efficiency in nutrient free medium after 48 h. These results indicate that, the clay/metaloxide/Ag could serve as efficient water purification technique, with a potential for large-scale commercialization.

## 1. Introduction

Pathogen-contaminated water source is a predominant problem in remote and rural communities, especially in poorer countries [1]. The communities in these regions remain poor and highly unskilled [2]. A simple to operate, efficient and cost effective approach is therefore paramount to solve the water treatment challenges in such communities. 

Metal and metal oxide nanoparticles are reported to offer water purification properties through their antibacterial activities and solutions at the point-of-use [3,4]. Meanwhile, the issue with nanoparticles is that nanoparticles contribute to the turbidity of the water sample. Some further dissolve in the water medium causing a drastic change in the chemical make-up of the water sample [5]. Thus, the introduction of nanoparticle powder into water sample causes a quick release of its ions in solution which becomes difficult to control. Hence the ionic release is highly unrestrained and it is equally difficult to predict the ionic time kinetic release [6]. Nanoparticle dissolution and their fast uncontrolled ionic release cause them to be used-up shortly. This phenomenon is responsible for the lack of the reuse of powdered nanoparticles for water purification [7]. Powdered nanoparticles in solution are also difficult to separate especially when they have undergone dissolution [8]. The solution therefore remains to control the release of ions from nanoparticles, prevent their possible dissolution, improve their reusability and enhance their separability in solution.

Membrane-based water purification systems provide an easy process for decontaminating water [9]. The technology ensures water reclamation through waste water treatment, removal of organic pollutants from brackish water and sea water desalination [10]. Current membrane-based treatment technologies include microfiltration, ultrafiltration, nanofiltration, and reverse and forward osmosis [11]. Clay material presents suitable properties for the design of a membranes system. Clay materials possess biocidal activities against a host of microbial species [12]. Polymeric materials implemented in a reverse osmosis process for water purification were susceptible to fouling [13]. However, processed clay materials have demonstrated antifouling property which is an important quality for membrane materials [14]. In addition to these, clay materials are naturally occurring predominant minerals that form a significant part of the Earth’s crust. Therefore, it is easy to acquire clay materials hence greatly reducing the cost of production as compared to polymer membranes [15]. Furthermore, the fabrication process of clay membranes is simple and is widely known and practiced across most cultures. This makes the use of clay suitable for adoption as technologies by people from developing countries that are highly unskilled.

In order to further enhance the biocidal activity of clay materials, antibacterial nanoparticles are incorporated into clay membranes [16]. These nanoparticles are mostly synthesized from metal or metal oxides or a nanocomposite comprising of the synergetic combinations of both or two or more of each particular type [17]. The antibacterial activities of silver, copper and zinc nanoparticles combined with clay have been previously explored [18]. However, a study of the integrated effect of a nanocomposite comprising the synergy of a metal oxide and a metal incorporated within clay minerals remains uninvestigated. Meanwhile, copper oxide/silver (CuO/Ag) and zinc oxide/silver (ZnO/Ag) nanocomposites have independently shown broad-spectrum and a highly efficient antibacterial activities relative to their respective separate metal oxide and metal nanoparticle counterparts [19,20]. Clay–nanocomposite creates a synergy of antibacterial activity from both clay and nanocomposite (nanoparticles) with dominating antibacterial efficacy exhibited by the nanoparticle agents [21]. Nanoparticles strongly bind to clay materials as fillers within clay matrices to form strong attachments in a solid composite. The material strength and thermal properties of the composite is enhanced through calcination at a temperature normally ranging from about 700 °C to 900 °C [21]. Other additive materials such as grog, sand, etc. strengthen it further to withstand dissolution in water [21]. Dissolution resistance allows the reuse of the clay ceramic materials for water purification. The composite may be in the shape of a small granule or pellet of a few millimeter size, cylindrical of a certain centimeter length among other suitable shape and desired sizes. Nanoparticles fillers within clay–nanocomposites release their toxic ions harmful to bacteria cells through the pores of the material [22].

In this study, bentonite clay locally acquired in Ghana and nanocomposite (copper oxide/silver (CuO/Ag) or zinc oxide/silver (ZnO/Ag)) were used for the fabrication of clay–nanocomposite pellets for water purification. Two separate clay–nanocomposite pellets were thus synthesized, made up of clay/CuO/Ag and clay/ZnO/Ag. Three (3) different samples were prepared for each type of clay–nanocomposite pellet containing varying concentrations of nanocomposite fillers (copper oxide/silver (CuO/Ag) or zinc oxide/silver (ZnO/Ag)) in the increasing order of 2.5, 5 and 10% (*w*/*v*), thus: clay/CuO/Ag had the following samples: clay/CuO/Ag-1, clay/CuO/Ag-2 and clay/CuO/Ag-3 while the samples of clay/ZnO/Ag were clay/ZnO/Ag-1, clay/ZnO/Ag-2 and clay/ZnO/Ag-3 pellets. These were then characterized and separately tested on clinical strain gram negative and gram positive bacterial species of *E. coli* (ATCC 25922) and *S. aureus* (ATCC 25923) respectively in a nutrient rich broth aqueous medium for 24 h and nutrient-free sterile aqueous medium for 24 and also 48 h using the colony forming unit (CFU) method for their antibacterial performance. The study aims to produce a cost-effective and efficient clay–nanocomposite for water purification at the point of water use.

## 2. Materials and Methods

### 2.1. Synthesis of Nanocomposites

The metal oxide/silver nanocomposites, i.e., CuO/Ag and ZnO/Ag were synthesized to function as fillers within clay material matrices for the fabrication of clay–nanocomposite pellets. They were deemed the principal antibacterial agents of the pellets. All chemical used for the synthesis of the metal oxide/metal nanocomposites were analytical grades hence did not need further purification. Copper (II) acetate dihydrate (Cu(CH_3_COO)_2_ × 2H_2_O, purity ≥99%), zinc acetate dihydrate (Zn(CH_3_COO)_2_ × 2H_2_O purity 99%), sodium hydroxide (NaOH, purity ≥ 99%), sodium borohydride (NaBH_4_ purity ≥ 99%), and silver nitrate (AgNO_3_, purity ≥ 99%) were obtained from Sigma Aldrich, Sent Louis, MO, USA.

A mixed-wet chemical method used for the nanocomposite synthesis is described in [19].

### 2.2. Synthesis of Clay–Nanocomposite

The materials used for the preparation of clay membranes for water purification which is given in Table 1 are bentonite clay material locally obtained in the western region of Ghana, metal oxide/metal nanocomposite, starch and polyvinyl alcohol (PVA). The PVA (9002-89-5) and starch (9005-84-9) were obtained from Sigma Aldrich, Sent Louis, MO, USA. These chemicals were analytical grade and did not require any further modification. The bentonite clay obtained as well as the synthesized nanoparticles were appropriately treated and characterized. Consequently, nanocomposites of copper oxide/silver (CuO/Ag) nanocomposite and zinc oxide/silver (ZnO/Ag) nanocomposite were also prepared. PVA was used as a binder in the clay nanocomposite preparation while starch was added for pore formation in the clay-nanoparticle composite pellets. The creation of pores in the membrane was achieved during calcination.

The clay nanocomposites were prepared by a facile admixture method of fabrication. The clay was treated via removal of roots and other combustible organic materials or impurities, dried in the sun to remove its moisture content. The clay was then ground to obtain a fine powder. The powered clay was sieved with a 90 µm mesh size. 20 g of the fine clay powder as shown in Table 1 was measured to a previously cleaned and sterile surface. To this, a fixed amount of starch (2.8 g), Table 1, was added and thoroughly mixed to obtain a homogeneous solid mixture. This was followed by the addition of a measured amount (given in Table 1) of respective nanocomposite (CuO/Ag or ZnO/Ag). The solid mixture was thoroughly mixed to ensure that the nanoparticles were uniformly dispersed within the clay material. A PVA solution was prepared by dissolving 1 g of PVA in 50 mL of distilled water and heating at 80 °C to ensure complete dissolution. This solution was then added to the solid clay-starch-nanoparticles composite mixture in minute quantities to achieve a plastic workable composite of clay–starch–nanoparticle. Spherical ball pellets of approximately 6 mm in diameter were made from the plastic composite. These pellets were dried in a desiccator for at least 48 h followed by drying in an oven at 110 °C. This was then followed by calcination at 800 °C for 4 h. The prepared ball pellets of clay-nanoparticle composite were ready as clay–nanocomposite pellets for water purification applications. Figure 1 is the annotated diagram for the synthesis of clay nanocomposite pellets.

### 2.3. Colony-Forming Units

The antibacterial potency of the clay nanocomposites were tested on strains of bacteria species of *E. coli* (ATCC25922) and *S. aureus* (ATCC25923). In the first instance, clay nanocomposite pellets were tested on bacteria species actively growing in a nutrient broth media and also when bacteria cells are suspended in a water medium without nutrient broth media but contained phosphate buffer saline (PBS) to maintain cellular integrity. The latter was to demonstrate a more real world case of bacteria in ordinary common water sources.

#### 2.3.1. Bacteria in a Nutrient Broth Media

The clay nanocomposites were tested on *E. coli* and *S. aureus* which are, respectively, Gram-negative and Gram-positive bacteria. These bacteria cells were respectively suspended in Luria–Bertani (LB) broth; 0.2 OD_600_ (optical density of wavelength of 600 cm^−1^) of bacteria cells were, respectively, measured using an ultraviolet–visible (UV–Vis) spectrophotometer. The 0.2 OD_600_ corresponds to 1.6 × 10^7^ CFU of respective bacteria cells. 1ml inoculum of the 0.2 OD_600_ were prepared in to sterile vials. 2 pellets of the individual prepared clay membranes were added to the same vials with respective bacteria inoculum. The bacteria inoculum was allowed to grow at 37 °C for a determined time of incubation of 24 h. The inoculums in respective vial were spread uniformly on a solid Mueller Hinton agar (MHA) in a disk plate. MHA was prior prepared by autoclaving at 121 °C for 15 min. The bacteria spread on MHA were allowed to incubate at 37 °C for overnight. The Bacteria colonies observed on the surface of MHA were then counted.

#### 2.3.2. Bacteria in a Sterile Water Medium

In the second antibacterial test of clay nanocomposites, the bacteria were not grown in a broth media but they were suspended in sterile water containing PBS. This experiment was carried out to observe the effects of the clay–nanocomposite pellets in purifying a more real case of bacteria infected water source devoid of active growing nutrients. To this, 1 mL of distilled water in a vial was contaminated with 0.2 OD_600_ of the two individual bacterial species with 2 pellets of separate clay–nanocomposite added to the respective vials. This was allowed to incubate at 37 °C for 24 h and then repeated for 48 h. The respective incubated bacteria were pipetted out and with the help of a sterile swap it was uniformly spread on an MHA in a disk plate. This was also incubated at 37 °C for overnight. The colonies formed were observed and counted for analysis of the antibacterial activity of the clay membranes.

## 3. Results and Discussion

### 3.1. Characterization of Metal Oxide/Silver Nanocomposite

#### 3.1.1. X-ray Diffraction (XRD) Analysis of Metal Oxide/Silver Nanocomposite

The synthesised metal oxide/silver nanocomposites were i.e., copper oxide/silver (CuO/Ag) and zinc oxide/silver (ZnO/Ag). The nanocomposites functioned as the principal antibacterial agent acting as fillers within clay matrixes for the formation of clay–nanocomposite pellets. The crystallinity of the dry nanocomposite powders were studied with an X-ray diffractometer (XRD) prior to the formation of clay–nanocomposite pellets by an admixture method. The XRD analysis was to study the crystalline microstructure of the respective nanocomposite samples. The XRD analysis (2θ scan) was performed at 40 kV and 30 mA with a CuKα radiation with a wavelength of 1.5406 Å within a range of 10° to 80°.

The crystal structure analysis of the CuO/Ag nanocomposite in Figure 2 revealed that the monoclinic phase of CuO was the dominant phase. The face-centered cubic (FCC) phase of silver (Ag) in the nanocomposite was weak, signifying the low percentage of silver in the nanocomposite. The crystalline nature of the powdered sample was confirmed by the sharpness of the XRD spectra. According to the Scherer equation, the crystal size of the CuO/Ag nanocomposite was found to be 34.2 nm. The nonappearance of unwanted peaks in the XRD spectral pattern pointed to the purity of the powered CuO/Ag nanocomposite sample which was further confirmed by X-ray fluorescence analysis.

The XRD spectral patterns of ZnO/Ag in Figure 3 showed exact coincidence with the standard patterns of the Joint Committee on Powder Diffraction (JCPD) for the wurtzite phase of ZnO. The dominance of the wurtzite phase of ZnO confirmed the high composition of ZnO in the ZnO/Ag nanocomposite. However, the peak occurring at 2θ of 38° corresponds to the 111 plane of silver. The weak spectra peaks of silver were indications of the low composition of silver in the ZnO/Ag nanocomposite. Silver had weak peaks due to its low concentration in the nanocomposite. A similar study have mentioned the weak spectra peaks of silver observed in composites with low silver content [23]. The Scherer equation revealed that the average size of the ZnO/Ag particle was 36.17 nm which was comparable to that of copper oxide [20].

#### 3.1.2. X-ray Fluorescence (XRF) Analysis of Metal Oxide/Silver Nanocomposite

The elemental analysis of the CuO/Ag and ZnO/Ag nanocomposites are respectively shown in Table 2. Chemical analyses of the nanocomposites were undertaken using a Spectro XLab 2000 X93-ray fluorescence (XRF) spectrometer (AMETEK, Berwyn, PA, USA). 4 g of dry powdered samples of equal quantities of the respective nanocomposite were subjected to XRF measurement. The detected constituents of the respective samples were expressed as relative percentages of oxides. Therefore, silver (Ag) was informed as Ag_2_O. The outcome of the XRF analysis affirmed that the respective nanocomposites were composed of similar constituents of respective metal oxides and silver.

#### 3.1.3. Transmission Electron Microscope (TEM) Analysis of Metal Oxide/Silver Nanocomposites

Transmission electron microscope (TEM, JEOL 2010, JEOL Ltd., Peabody, MA, USA) micrographs of the respective metal oxide/silver nanocomposites are given in Figure 4 and Figure 5. TEM images were obtained by coating the surface of a copper grid with a drop of metal oxide/silver suspension. The dry copper grid was used for TEM analysis. The micrographs were further analyzed with ImageJ (National Institute of Health and the Laboratory for Optical and Computational Instrumentation). The statistical distribution of the respective nanocomposite particle size was thus obtained. CuO/Ag nanocomposites in Figure 4 were rod-shaped with a length ranging from 200 to 900 nm with 400 nm as the modal length. The width of the CuO/Ag nanocomposite particles ranged from 5 nm to 25 nm with a modal width of 20 nm. The difference in particle size recorded among CuO/Ag samples by TEM and XRD was attributed to the possibilities of agglomeration. In the past a broad range of bacteria strains have recorded susceptibility to CuO particles of similar features [20].

The mixed wet-chemical synthesized ZnO/Ag nanocomposites assumed plate-like morphologies similar to that already reported by Jang et. al. [19]. The ZnO/Ag particles in Figure 5 recorded a size distribution ranging from 15 nm to 60 nm. The average particle size was 35 nm. The particle size recorded by the TEM analysis was, however, similar to that obtained by the Scherer’s equation. Similar particle sizes have demonstrated activity against bacteria strains [19,24]. The Scherer’s equation revealed that CuO/Ag and ZnO/Ag particles possessed a similar particle size of 34.2 nm and 36.17 nm, respectively. The antibacterial activity of nanoparticles is highly dependent on particle size. The smaller the particle size the higher its corresponding antibacterial activity. Particle size influences antibacterial activity through a number of mechanisms especially reactivity. The number of atoms on particle surface increases with decreasing size. The surface atoms engage in a chemical reaction with the bacterial cells leading to cell death. Similarly, the particle size also affects the internalization of particles leading to cell death. Therefore, similar particle sizes are expected to exhibit a relatively comparably antibacterial activity. 

#### 3.1.4. Ultraviolet–Visible (UV–Vis) Spectroscopy Analysis of Metal Oxide/Silver Nanocomposites

In Figure 6, the UV-Vis (Genesys 150 UV–Vis, ThermoFisher Scientific, Waltham, MA, USA) spectrophotometric absorption spectrum for CuO/Ag nanocomposite particles is shown. 1 mL of CuO/Ag aqueous suspension in a quartz curvet with the same measure of distilled water as the blank was used for the UV–Vis measurement. The peak wavelength of the CuO/Ag particles was observed to be 542 nm. The energy wavelength for electron transmission from the valence band to the conduction is known to occur at the peak wavelength [25]. Bandgaps of the respective nanocomposites were obtained from the Tauc plot by extrapolating the line on the curve to the corresponding x-axis (*hν*).

The plot revealed that CuO/Ag had a bandgap of 1.2 eV. Previously CuO was shown to have a band gap of 1.5 eV [25]. The deviation in our calculated bandgap was attributed to the fact that the presence of silver in the nanocomposite caused an enhancement of the band gap through its doping effects. Thus, the silver acted as a dopant for the CuO thereby enhancing the band gap of CuO/Ag. The band gap of the material affects its respective electrical conductivity hence the release of electrons. The lower the band gap the higher it’s associated conductivity. The release of electrons by the nanocomposite in solution consequently results in the generation of reactive oxygen species (ROS) which are deadly to bacteria cells. The UV–Vis of CuO/Ag had an extended peak from 420 nm to 750 nm which lies within the visible light energy range. Visible light could therefore cause electron propagation from the valence band to the conduction band which is responsible for causing ROS leading to bacterial cell death.

Figure 7 is the UV–Vis of ZnO/Ag nanocomposite. ZnO/Ag showed highest peak of UV-Vis spectral absorption at 372 nm. Therefore, electron transmission from the valence band of ZnO/Ag to its conduction was predominant at the energy wavelength of 372 nm, which falls within the ultraviolet light energy range. Thus, the release of electrons from ZnO/Ag heavily occurred within the UV range. ZnO/Ag particles required ultraviolet light for electron generation which interacts with bacterial cells resulting in ROS activities and subsequent cell death. Our calculated wavelength for ZnO/Ag agrees with earlier work on the wavelength of ZnO nanoparticles [20]. The energy band gap of ZnO/Ag was found to be 2.88 eV. Most researchers have calculated the energy band gap of ZnO to be 3.2 eV. The enhanced energy band gap of ZnO/Ag as compared to ZnO was due to the fact that silver introduces a doping effect in ZnO particles, thereby resulting in an enhanced energy band gap of ZnO/Ag. ZnO/Ag therefore required lesser energy to break its band gap for electron transfer from the valence band to the conduction band leading to ROS generation and eventual bacteria cell death [26].

CuO/Ag had the highest peak of UV–Vis spectral absorption in the lower energy band of visible light as compared to ZnO/Ag which was observed within the relatively high energy band of the ultraviolet range. Therefore, CuO/Ag required lesser energy for electron transition between energy states as compared to ZnO/Ag. Thus CuO/Ag could produce electrons for ROS activities at a lower energy as compared to ZnO/Ag particles. This observation is also true about the bang gaps of the two particle types. The band gap of CuO/Ag was 1.2 eV while that of ZnO/Ag was 2.88 eV.

### 3.2. Clay–Nanocomposite Pellets

The synthesized clay–nanocomposite pellets which were visible to the unaided eye are demonstrated in Figure 8. The pellets were porous spherical in shape having a diameter of about 6 mm and weighing approximately 0.04 g (40 mg ± 10) per pellet. There were three (3) different kinds of each type of pellet (clay/CuO/Ag or clay/ZnO/Ag) according to the concentration of nanocomposite (CuO/Ag/ or ZnO/Ag) fillers. They were clay/CuO/Ag-1, clay/CuO/Ag-2, clay/CuO/Ag-3, clay/ZnO/Ag-1, clay/ZnO/Ag-2 and clay/ZnO/Ag-3. (1) Contained 2.5% of nanocomposite as fillers, (2) contained 5% of nanocomposite as fillers and (3) contained 10% of nanocomposite as fillers.

The porous structure was created during calcination at 800 °C by the decomposition of starch which is incorporated in the fabrication of the material as a binder. The amount of starch and clay material in each type of pellet was constant at 2.8 g. A porous structure is important for the free diffusion of ions form the pellet structure into the aqueous media to mediate the cell death of bacteria species. In Figure 9, samples of both CuO/Ag and ZnO/Ag clay–nanocomposite pellets are shown.

#### 3.2.1. XRD Analysis of Clay–Nanocomposites

The XRD spectra patterns of clay mineral, modified clay materials and CuO/Ag nanocomposite which were the clay modifiers is given in Figure 9. The XRD spectra peak at diffraction angle, 2θ = 20.90, 26.80, 36.84, 39.43, 40.40, 42.61, 50.18, 55.22, 60.17, 68.44 which matches with the following lattice planes: (100), (012), (110), (104), (112), (200), (114), (016), (122) and (032). These spectra are a representation of the presence of clay minerals. Modified clay samples: clay/CuO/Ag-1, clay/CuO/Ag-2 and clay/CuO/Ag-3 contain 2.5%, 5% and 10% respective percentages of CuO/Ag. Clay/CuO/Ag-1, clay/CuO/Ag-2 and clay/CuO/Ag-3 show spectra peaks which corresponds with CuO/Ag. This indicates the presence of CuO/Ag in the modified samples.

Similarly Figure 10 has samples of clay, modified clay and ZnO/Ag nanocomposites as clay modifiers. The modified clay samples: clay/ZnO/Ag-1, clay/ZnO/Ag-2, clay/ZnO/Ag-3 accordingly contained 2.5%, 5% and 10% of ZnO/Ag. The XRD patterns therefore accordingly contained spectra peaks corresponding to clay mineral. Similarly these peaks were identified at the following diffraction angles, 2θ = 20.90, 26.80, 36.84, 39.43, 40.40, 42.61, 50.18, 55.22, 60.17 and 68.44 which corresponds with the following lattice planes: (100), (012), (110), (104), (112), (200), (114), (016), (122) and (032). The modified clay samples: clay/ZnO/Ag-1, clay/ZnO/Ag-2 and clay/ZnO/Ag-3 similarly contained 2.5%, 5% and 10% corresponding percentages of ZnO/Ag. Corresponding spectra peaks of ZnO/Ag are identified in the modified clay samples of clay/ZnO/Ag-1, clay/ZnO/Ag-2 and clay/ZnO/Ag-3. The weak peaks of ZnO/Ag in the modified clay samples’ spectra patterns is due to the low concentration of ZnO/Ag in the modified clay samples.

#### 3.2.2. X-ray Fluorescence (XRF) Analysis of Clay–Nanocomposites

The XRF (Bruker, Billerica, MA, USA) chemical analysis of clay/CuO/Ag-1, clay/CuO/Ag-2 and clay/CuO/Ag-3 are in Table 3 while Table 4 has the following: clay/ZnO/Ag-1, clay/ZnO/Ag-2 and clay/ZnO/Ag-3 clay–nanocomposite samples. XRF chemical analysis revealed that SiO_2_ and Al_2_O_3_ were the respective principal constituents of all clay–nanocomposite samples. This confirms the presence of aluminosilicate clay minerals in the samples. The silica composition of samples ranged from 51.8% to 65.4% and the alumina component ranged from 16.4% to 19.9%. Thus, the silica to alumina ratio ranged from 3.28 to 3.15. A high silica content and a relative low alumina content suggest that the clay minerals have an octahedral sheet which normally contains central aluminium sandwiched between two tetrahedral silicate sheets. This type of clay lattice parameter is associated with a bentonite clay type. The XRF analysis also revealed the presence of other oxides of cations including Mg^+^, Mn^+^, K^+^, Ca^+^, Ti^2+^, Fe^3+^ and Na^+^. The cations identified in the samples suggest that the central octahedral ions may include: Al^3+^, Mg^2+^ or Fe^3+^. The various cations also suggest that the clay mineral has charged surfaces that provide it with high ion-exchange capability. High Ion exchange capability is particularly unique of bentonite clays [27]. Ion exchange capability provides special property to clay minerals such as adsorption of charged elements and molecules from sources such as water. This would be an added advantage to that of antibacterial activity of the clay–nanocomposite pellets.

In Table 3, clay–nanocomposite pellet samples: clay/CuO/Ag-1, clay/CuO/Ag-2 and clay/CuO/Ag-3 recorded respective increasing amounts of CuO and Ag in accordance of CuO/Ag used in their respective synthesis. All elements were expressed in oxides hence Ag was expressed as Ag_2_O. The low relative percentage of CuO/Ag recorded by XRF agrees with the low CuO/Ag XRD spectra peaks.

In Table 4, the components of clay/ZnO/Ag samples: clay/ZnO/Ag-1, clay/ZnO/Ag-2 and clay/ZnO/Ag-3 are indicated. The samples showed corresponding increase in ZnO and Ag which agrees accordingly with the relative amount of ZnO/Ag incorporated in each sample. All samples were expressed in relative percentages of oxide including silver. Hence, silver was expressed as Ag_2_O.

#### 3.2.3. Scanning Electron Microscope (SEM) Analysis of Clay–Nanocomposites

Figure 11 shows the scanning electron microscope (SEM, Bruker, Billerica, MA, USA) micrographs of the inside of clay–nanocomposite pellets of clay/CuO/Ag-1, clay/CuO/Ag-2 and clay/CuO/Ag-3. The CuO/Ag nanocomposite particles are observed to be uniformly distributed on the clay surfaces. This means the clay–nanocomposite synthesis method was able to achieve a uniform dispersion of CuO/Ag within clay matrixes. The synthesis processes included the thorough mixing of clay with CuO/Ag. Clay/CuO/Ag-1, clay/CuO/Ag-2 and clay/CuO/Ag-3 contained respective increasing amounts of CuO/Ag. The clay in the micrograph are dense and highly compact which is attributed to the calcination at 800 °C. The CuO/Ag nanocomposites are seen on the clay surfaces.

Figure 12 has the SEM micrographs of clay–nanocomposites of clay/ZnO/Ag samples. clay/ZnO/Ag-1, clay/ZnO/Ag-2 and clay/ZnO/Ag-3 contain increasing amounts of ZnO/Ag nanocomposite fillers which is confirmed in the SEM micrographs. Flakes of ZnO/Ag particles are observed on the surfaces of dense clay material. The SEM image of the cross section of clay–nanocomposite pellets shows that the ZnO/Ag nanocomposite fillers were uniformly dispersed within the bulk clay matrixes.

Metal oxide/silver (CuO/Ag and ZnO/Ag) nanocomposite are the main antibacterial agent within clay–nanocomposite pellets. Uniform dispersion of nanocomposite with clay pellets ensures that the pellets have uniform antibacteria active sites on its surfaces.

#### 3.2.4. UV–Vis Spectrophotometer of Clay–Nanocomposites

The UV–Vis (Genesys 150 UV–Vis, ThermoFisher Scientific, Waltham, MA, USA) absorption spectra of clay/CuO/Ag-1, clay/CuO/Ag-2 and clay/CuO/Ag-3 are shown in Figure 13. The samples had a broad spectral peak from 390 nm to 550 nm. The samples have absorption largely within the range of visible light wavelenght. Electron conduction band to the valence band predominantly occurred within the wavelenght of visible light. The visible light wavelength of energy is responsible for the generation of reactive species. Reactive species such as hydrogen peroxide, superoxide anion, peroxides and hydroxyl radicals interact with the surfaces of bacteria cell resulting cell death.

The energy band gap of the samples are also computed from their UV–Vis absorption spectra. From Figure 13B, the energy band gaps of samples were in the following descending order: clay/CuO/Ag-1 > clay/CuO/Ag-2 > clay/CuO/Ag-3. The band gap decreased with the increase in the composition of CuO/Ag in the samples. The content of CuO/Ag in the samples were in the ascending order of: clay/CuO/Ag-1 < clay/CuO/Ag-2 < clay/CuO/Ag-3 with the respective band gaps of 1.80 eV > 1.46 eV > 1.42 eV. Increase in the content of CuO/Ag in the samples influenced their band gaps to approach the band gap of CuO/Ag which was recorded to 1.2 eV. Thus, more properties of CuO/Ag were introduced in the clay–nanocomposite samples.

Increase in the composition of CuO/Ag in the samples introduces a wide spectra peak which is seen in CuO/Ag-2 and CuO/Ag-3. The wide peak spectra mean that electron transmission between the discrete energy states can take place within an extended range of energy wavelenght. The wavelength of transition is mostly observed in the wavelength of visible light. The incidence of visible light on the sample allows the absorption of equivalent energy wavelength to initiate the production of charged species. Charged species interact with bacterial cells in multiple ways including: cell wall disruption, cellular leakage with subsequent attack on organelles, proteins, lipids and bacterial genomes. The energy of absorption increases with decreasing the wavelenght on absorption. Longer wavelength of absorption such as in the visible light range require lower energy for restive oxygen species generation than electromagnetic waves of shorter wavelenght such as ultraviolet light. The longer wavelenght that a sample possess the less energy demanded for producing reactive species. Hence, less energy is required for its antibacteria activity.

Figure 14 is the UV–Vis of clay/ZnO/Ag samples: clay/ZnO/Ag-1, clay/ZnO/Ag-2 and clay/ZnO/Ag-3. The samples had a broad spectra peak ranging from 380 nm to 570 nm. Thus the absorption wavelenght of the samples were significant within the visible light range. The composition of ZnO/Ag in the samples was increased in the order of: clay/ZnO/Ag-1, clay/ZnO/Ag-2 and clay/ZnO/Ag-3. However, unlike clay/CuO/Ag samples, there was no effect on the spectral peak. There was no further extension of the spectra peak. Clay/ZnO/Ag-2 showed an absorption peak at 380 nm which is characteristic of ZnO/Ag nanocomposite. The peak confirms the presence of ZnO/Ag in the clay–nanocomposite sample.

Increase in the amount of ZnO/Ag in samples impacted the band gap of samples. The band gap increased with increasing the ZnO/Ag content. It is assumed that the band gap approaches the band gap of ZnO/Ag as its content increases in the sample.

Comparatively, clay/CuO/Ag samples recorded lower band gaps that clay/ZnO/Ag samples. This means that the energy required for electron transition from the conduction band to the valence band of clay/CuO/Ag samples is lower than clay/ZnO/Ag samples. Clay/CuO/Ag samples would, therefore, produce reactive oxygen species at lower energy than clay/ZnO/Ag samples. Clay/CuO/Ag samples would therefore have a higher antibacteria activity at a constant lower energy such as visible light than clay/CuO/Ag samples. The production of reactive oxygen species are known as causes of bacterial cell death.

#### 3.2.5. Brunauer–Emmett–Teller (BET) Analysis of Clay-Nanocomposite Pellets

Brunauer–Emmett–Teller (BET, BEL, Osaka, Japan) analysis of the clay–nanocomposite pallets were undertaken to study the influences of starch which was included in the synthesis of the pellets for the creation of porous structure. Porous structure is important for the diffusion of metal ions and charged species from internal pellet structure into aqueous medium for antibacteria activity. All pellet samples in the study contained same amount of starch for pore creation. During calcination of pellet samples, the starch is burnt to create the pores within the clay structure. Metal ions and charged species can be released through pores or from the surfaces of pellets especially where metal oxide/metal (CuO/Ag or ZnO/Ag) nanocomposite resides. The BET analysis provided information on the surface area and porosity using nitrogen-gas adsorption.

Figure 15 contains the adsorption-desorption curves of clay–nanocomposites samples of clay/CuO/Ag-1, clay/CuO/Ag-2 and clay/CuO/Ag-3. Nitrogen gas-adsorption was measured as a function of relative adsorption pressure (*P/P_0_*) from 0.05 to 1. At (*P/P_0_* = 1) all the nitrogen gas is condensed inside the pores. Desorption was carried out by reducing the pressure from “1” which was the condensation pressure. The adsorbed gas was thus removed for the pores and the surfaces of pellets. The information acquired from this was used to obtain the adsorption-desorption isotherm of clay/CuO/Ag samples. The adsorption desorption isotherms were classified as Type IV isotherms according to the international union of pure and applied chemistry, IUPAC 1985 classification [28]. The isotherms provide qualitative information on the porosity of clay/CuO/Ag samples. According to the Type IV isotherms obtained, which are characteristic of pore sizes between 1.5 and 100 nm, these are classical signs of mesopores. The micropores were filled at very low relative pressure followed by the formation of a monolayer at the knee of the curve also at low relative pressure. At a moderate pressure, a multilayer of nitrogen gas molecules is formed and then as the pressure approaches a peak value there is condensation of nitrogen gas molecules in the pores (capillary condensation). The isotherm depicts H3 hysteresis type which is an indication that the pores have a slit shape. Gas adsorption is continuous at high *P/P_0_* which confirms the agglomeration of particles observed in the SEM analysis.

The specific surface area (S_BET_), pore volume and pore size distribution of the sample were computed from the nitrogen gas-adsorption. The specific surface area, pore volume and pore size are calculated by the implementation of the BET theory.

Figure 16 and Table 5 portraits information on the porosity and surface area of clay/CuO/Ag samples. Clay/CuO/Ag-1, clay/CuO/Ag-2 and clay/CuO/Ag-3 had peak pore sizes of 76.49, 90.22 and 103.54, respectively. However, all samples had a predominant number of micro and mesopores which are obvious in Figure 16. Pore volumes and surface area of samples also increased accordingly with increase in CuO/Ag fillers in clay. The phenomena proves the uniform dispersity of CuO/Ag within clay matrixes. In coincidence with the features of Type IV isotherms the samples had peak pore size near 100 nm.

The adsorption-desorption isotherms of clay/ZnO/Ag are shown in Figure 17. The samples similar to clay/CuO/Ag samples exhibited Type IV isotherms. This attests to the fact that the samples had related surface properties, hence behave identically. The similar characteristics observed can be attributed to the dominant proportion of clay in both samples and the similar fabrication techniques used for both of them.

The micropores of the samples were occupied with nitrogen gas molecules at an extremely low pressure. This was subsequently followed by monolayer coverage which begun at the knee of the isotherm curves and then multilayer formation at an average pressure and finally capillary condensation at the condensation pressure of “1”. Similarly, the pores were of the slit type and the sample showed aggregation features by the unlimited adsorption at high pressure.

Clay/ZnO/Ag samples in Figure 18 and Table 5 above demonstrates increase in surface area, pore volume and peak pore size with increase in ZnO/Ag filler content accordingly as: clay/ZnO/Ag-1, clay/ZnO/Ag-2, clay/ZnO/Ag-3. Clay/CuO/Ag samples followed the same pattern which agrees with their similar surface properties. It also attests to the fact that all samples were thoroughly mixed and achieved a uniform dispersity of nanocomposite fillers within clay matrixes.

Furthermore, in Table 5, clay/ZnO/Ag samples recorded higher values of surface area, pore volume and peak pore sizes than clay/CuO/Ag samples. This may attributed to the fact ZnO/Ag were more porous than CuO/Ag nanocomposites. This agrees with BET analysis reported by Elkady which revealed that zinc oxide nanoparticles subjected to different calcination temperatures could achieve a wide range of porosity ranging from 278.6 to 53 nm [29].

## 4. Antibacterial Activity of Clay–Nanocomposites

The colony forming unit (CFU) was the method adopted for testing the antibacterial activity of the clay–nanocomposite pellets. The pellets were tested on *E. coli* strain (ATCC25923) and *S. aureus* strain (ATCC25923) which are respectively Gram-negative and Gram-positive bacteria. The clay–nanocomposite pellets were composed of clay with nanocomposites (CuO/Ag or ZnO/Ag) fillers. The clay–nanocomposites prepared for antibacteria testing were clay/CuO/Ag and clay/ZnO/Ag. To test the effects of the pellets, different concentrations of nanocomposite fillers were used for the pellet formation. The concentrations were 2.5%, 5% and 10% which were respectively contained in clay/CuO/Ag-1, clay/CuO/Ag-2 and clay/CuO/Ag-3. The same concentrations were respectively contained in clay/ZnO/Ag samples including clay/ZnO/Ag-1, clay/ZnO/Ag-2, and clay/CuO/Ag-3. The pellets were tested on bacteria strains in liquid broth media and also in sterile water with phosphate buffer saline (PBS). PBS is introduced to maintain cellular integrity within water.

### 4.1. Antibacterial Activity of Clay–Nanocomposite Pellets on Bacteria in Broth Media

The colony forming units of the antibacterial activity of the clay–nanocomposite pellets were performed against the *E. coli* and *S. aureus* strains in an aqueous Mueller Hinton broth after 24 h of incubation. CFU testing after growth in a broth media is important to know the effective performance of the antibacterial agents on bacteria growing in an active nutrient source.

Figure 19 shows the effect of clay–nanocomposite pellets on *E. coli.* All the different pellets of clay/CuO/Ag including clay/CuO/Ag-1, clay/CuO/Ag-2 and clay/CuO/Ag-3 which contain respective increasing trends of CuO/Ag showed increasing activity on the *E. coli* strain accordingly. However, clay/ZnO/Ag pellets did not record activity on the same *E. coli* strain.

The activity of the clay–nanocomposite pellets against the *S. aureus* strain in shown in Figure 20. Just as observed among the *E. coli* strain, the *S. aureus* strains were susceptible to the clay/CuO/Ag pellets. The increasing concentrations of CuO/Ag respectively in the pellets: clay/CuO/Ag-1, clay/CuO/Ag-2 and clay/CuO/Ag-3 had a corresponding consistent trend of increase in their respective antibacteria activities against *S. aureus*. While clay/CuO/Ag-1 showed the relative lowest antibacterial activity, clay/CuO/Ag-3 showed the highest antibacteria against *S. aureus* in broth media. This observation was made among clay/CuO/Ag against *E. coli*. This means that the antibacterial activity was primarily due to the concentration of CuO/Ag in the pellet since the concentration of CuO/Ag was in the increasing order of: clay/CuO/Ag-1< clay/CuO/Ag-2 < clay/CuO/Ag-3. It can be deduced that the generation of metal ions were the main causes of cell death increased in the order of increasing CuO/Ag concentration in pellets. Apart from the self-toxic nature of metal ions to cells, metal ions can initiate reactive oxygen species which also cause cell damage.

The *S. aureus* strain was, however, not susceptible to clay/ZnO/Ag pellets. The different pellets following the trends of increasing concentration of ZnO/Ag: clay/ZnO/Ag-1 < clay/ZnO/Ag-2 < clay/ZnO/Ag-3 did not show activity against *S. aureus*. This is attributed to the quantity of metal ions produced in the solution by ZnO/Ag in the respective pellets. In the presence of oxygen, metal ions initiate oxidative stress. In the absence of oxygen. Cytoplasmic intracellular metal ions can also mediate cell death. Transition metal ions especially Fe^3+,2+^ and Zn^2+^ are essential micronutrients utilized for a number of cellular biological processes such as being important components of metalloproteins. Their toxicity therefore depends on their quantitative amount in the cellar environment. Quantities below harmful limits can be utilized by cell for metabolic processes or effectively controlled from hampering cellular activities through the cell’s control of its uptake and efflux systems. If the metal ions are not in excess but within limit range, the cellular uptake remains in touch and not disrupted. However, an excess metal ion unsettles the ion uptake system allowing excess ions into the cell causing intracellular damage. It is therefore assumed that ZnO/Ag where in low quantity to illicit disruptive mechanism to the cell. Again, within the environment of nutrient broth media, cells possess conducive atmosphere for enhanced growth and reproduction. Thus, the cells multiplication rate could be higher than any inhibition by the clay/ZnO/Ag sample pellets. As the multiplication rate of the cell overpowers the inhibition caused by clay/ZnO/Ag, the inorganic antibacteria agent becomes limited and its potency declines by the increasing bacteria population.

Clay/CuO/Ag samples of clay/CuO/Ag-1, clay/CuO/Ag-2 and clay/CuO/Ag-3 had respective energy band gaps of 1.80 eV, 1.46 eV and 1.42 eV which were lower than the energy band gaps of their counterpart clay/ZnO/Ag samples of clay/ZnO/Ag-1 clay/CuO/Ag-2 and clay/CuO/Ag which also had respective energy band gaps of 1.87 eV, 1.87 eV and 1.91 eV. The corresponding lower energy band gaps of clay/CuO/Ag samples means that they required lesser energy for electron transition across energy levels to generate reactive oxygen species responsible for antibacteria activities. However, clay/ZnO/Ag-1 clay/ZnO/Ag-2 and clay/ZnO/Ag-3 with a lower band gap than clay/CuO/Ag-1 still produced no antibacteria activity while clay/CuO/Ag-1 was effective against the bacteria. This means that the antibacteria activity was not only due to the requisite energy to overcome the band gap barrier. Zinc is second to iron as the most utilized transition metal ion in cellular metabolic processes. Zinc is much more required by the biological system than copper. The concentration of zinc in *E. coli* was found to be 10^−4^ M, 10^−6^ M and 10^−8^ M for intracellular and extracellular copper respectively. Under similar conditions, copper ions would be less used hence being in excess which possess toxic to cells [30]. Copper ions are highly affinitive towards metalloproteins. They form thiolate bonds comprising of iron-sulphur clusters. The action degrades the cluster and releases free iron which initiates the generation of cellular scavenging oxidative species [31]. Copper ions are highly reactive and generate reactive oxygen species by their participation in redox reactions as cupric and cuprous ions. Clay/CuO/Ag pellets could therefore greatly inhibit cellular activities.

Figure 21 is a histogram comparing the antibacteria activity of clay/CuO/Ag and clay/ZnO/Ag pellets against *E. coli* and *S. aureus*. The findings illustrated by the histogram reveals that both *E. coli* and *S. aureus* were insensitive to clay/ZnO/Ag pellet samples. Clay/ZnO/Ag samples of clay/ZnO/Ag-1 < clay/ZnO/Ag-2 < clay/ZnO/Ag-3, in increasing concentrations of ZnO/Ag nanocomposite had a zero (0%) activity on the inhibition of both bacteria species in nutrient rich broth media. However, clay/CuO/Ag pellets, in accordance with the increasing concentration of CuO/Ag within clay, thus clay/CuO/Ag-1< clay/CuO/Ag-2< clay/CuO/Ag-3 showed increasing activity against *E. coli* as 20%, 26% and 31%. Similarly, clay/CuO/Ag-1 < clay/CuO/Ag-2 < clay/CuO/Ag-3 showed 26%, 32% and 40% activity against *S. aureus*. The result agrees with earlier findings of previous study by Lesley and co-workers [31] where CuO recorded higher antibacteria activity as compared to ZnO.

### 4.2. Antibacteria Activity of Clay–Nanocomposite Pellets after Incubation in Water

The CFU of antibacteria activity of clay–nanocomposite pellets were also performed on bacteria strains in double distilled sterile water without an active nutrient broth media after 24 h of incubation and also after 48 h of incubation. Bacteria are common contaminants of untreated water which causes diseases and mortality especially among infants and adults with compromised immunity. The investigation will enlighten understanding of clay–nanocomposites for water purification of regular water sources such as surface waters, underground waters and other possible contaminated sources.

All the pellets of clay/CuO/Ag including clay/CuO/Ag-1, clay/CuO/Ag-2 and clay/CuO/Ag-3 exhibited activity against the *E. coli* strain. clay/CuO/Ag-1 had 96.4% activity on the death of *E. coli* cells. Clay/CuO/Ag-2 and clay/CuO/Ag-3 respectively resulted in the total inhibition of *E. coli* cells. They recorded 99% death of E. coli cells. Among clay/ZnO/Ag pellets, clay/ZnO/Ag-1 and clay/ZnO/Ag-2 pellets had minimal effect on the inhibition of *E. coli* cells. Meanwhile, clay/ZnO/Ag-3 had a 100% efficiency in the inhibition of *E. coli* cells. Clay/ZnO/Ag-3 had a higher composition of ZnO/Ag as compared to clay/ZnO/Ag-1 and clay/ZnO/Ag-2. Figure 22 containing the improved activity recorded in clay/ZnO/Ag-3 is attributed to its higher content of ZnO/Ag.

The activity of clay–nanocomposite pellets against *S. aureus* are found in Figure 23. The clay/CuO/Ag-1, clay/CuO/Ag-2 and clay/CuO/Ag-3 samples were all efficient antibacteria agents against *S. aureus*. Clay/CuO/Ag-1, clay/CuO/Ag-2 respectively showed 99.4% and 99.8% efficiency in inhibiting the growth of *S. aureus* cells in sterile water while clay/CuO/Ag-3 was capable of a 100% inhibition. Clay/ZnO/Ag-1and clay/ZnO/Ag-2 samples recorded negligible activity against the *S. aureus* strains. The *S. aureus* cells were however more sensitive to clay/ZnO/Ag-3 pellets. It recorded a complete removal of the cells by causing a 100% inhibition of the cells. The results is attributed to the concentration of ZnO/Ag in the clay/ZnO/Ag-3 sample which was higher than clay-ZnO/Ag-1 and clay/ZnO/Ag-2 samples. Clay/ZnO/Ag-3 was deemed to generate excess Zn^2+^ and Ag^+^ which influenced cellular activity causing cell death. Unlike broth nutrient rich medium, the bacteria medium in this case contained no active nutrient source to keenly promote bacteria growth hence they could be inhibited by the right amount of antibacteria agents. The lack of activity shown by clay/ZnO/Ag-1 and clay/ZnO/Ag-2 is principally due to the quantity of Zn^2+^ and Ag^+^ which they produced in the bacteria environment. Cell deaths caused by metal ions are almost impossible unless they are in excess quantity within the cellular environment. Otherwise, they are utilized for cellular metabolic activities or effectively controlled cell mechanisms and prevented from any meaning effect causing cell death. Metal ions are trace elements serving as cofactors and as components for the structure of enzymes.

Figure 24 relates the antibacteria activity of clay/CuO/Ag and clay/ZnO/Ag against *E. coli* and *S. aureus*. The results indicate that, clay/CuO/Ag pellet samples had a magnificent activity against both bacteria species after 24 h of incubation in sterile water. The activities of clay/CuO/Ag samples against *E. coli* strain were 96.69%, 99.4% and 99.75% respectively by clay/CuO/Ag-1, clay/CuO/Ag-2, and clay/CuO/Ag-3. Again, the activities against *S. aureus* were 99.44%, 99.82 and 100% by clay/CuO/Ag-1, clay/CuO/Ag-2, and clay/CuO/Ag-3 respectively. While clay/ZnO/Ag-1 and clay/ZnO/Ag-2 demonstrated no (0%) activity against both *E. coli* and *S. aureus*, clay/ZnO/Ag-3 recorded a tremendous high antibacteria activity as 100% against both bacteria species. The results obtained shows that the antibacteria activity was dependent on the concentration of nanocomposite (CuO/Ag or ZnO/Ag) contained in the clay matrixes. Clay/ZnO/Ag-3 with a higher concentration of ZnO/Ag than clay/ZnO/Ag-1 and clay/CuO/Ag-2 was able to inhibit bacteria contaminants unlike clay/ZnO/Ag-1 and clay/ZnO/Ag-2 which had a relatively lower amount of ZnO/Ag. The results obtained according to the concentration of nanocomposites in pellet samples agrees with that of others where the activity against *E. coli* and *S. aureus* were dependent on the respective concentrations of CuO, ZnO, CuO/Ag and ZnO/Ag [32].

The antibacteria activity of clay–nanocomposite pellets obtained also shows that the antibacteria activity of the pellets was dependent on the environment on the bacteria species. With respect to the results obtained, clay/ZnO/Ag-3 pellets had no activity on both bacteria species in broth media. Meanwhile without broth media and maintaining the time of antibacteria activity, clay/ZnO/Ag-3 showed a vast improvement in its antibacteria activity against both bacteria species which proves that the antibacteria activity of the pellets was affected by the bacteria environment. While the activity of the pellet was almost insignificant in broth nutrient rich media environment it was high effective in an environment without nutrient. The increased antibacteria activity of clay/CuO/Ag-1, clay/CuO/Ag-2 and clay/CuO/Ag-3 in nutrient free environment as compared to the nutrient rich broth media environment again demonstrates that the antibacteria activity of the clay–nanocomposite pellets depended on the bacteria environment. The environment affects the potency of the antibacteria agents. Higher antibacteria potency is expected in nutrient rich environments supporting bacteria growth. The results of which is seen in the difference in activity between clay/CuO/Ag pellet samples and clay/ZnO/Ag pellet samples in nutrient rich broth media and sterile water without nutrient.

The CFU of antibacteria activity of clay–nanocomposite pellets recorded after 48 h incubation of bacteria cells in water was undertaken to study the influence of time on the antibacteria activity of the clay–nanocomposite. Figure 25 shows the results of clay/CuO/Ag-1, clay/CuO/Ag-2, clay/CuO/Ag-3, clay/ZnO/Ag-1, clay/ZnO/Ag-2 and clay/ZnO/Ag-3 on *E. coli*. The results shows the samples were effective in the inhibition of the *E. coli* strain. Clay/CuO/Ag-1 had a 99.94% efficiency in the inhibition of *E. coli* while clay/CuO/Ag-2 and clay/CuO/Ag-3 had a 100% efficiency. Clay/ZnO/Ag-1 and clay/ZnO/Ag-2 could inhibit the *E. coli* growth by 99.71 and 99.85% respectively. However, clay/ZnO/Ag-3 achieved a 100% removal of *E. coli* from the sterile water medium.

The activity of clay–nanocomposites pellets against *S. aureus* after 48 h of incubation in sterile water is recorded in Figure 26 below. All the samples tested on the bacteria species for 48 h of incubation had an extremely high activity on the *S. aureus* strain. 48 h allows an extensive time for the production of respective ions in the water medium. The antibacteria activity is known to be dependent on the concentration of present ionic species which attacks bacteria cells. Overwhelming amount of ions in the cellular cytoplasm overpowers the cells ability to remove them through cellular efflux and other cell defence mechanisms. The ions cause cell stress, genetic alterations, changes in organelle structures, cell wall disruptions among others.

Clay/CuO/Ag-1, clay/CuO/Ag-2 and clay/CuO/Ag-3 samples achieved a 100% efficiency in the removal of *S. aureus*. Clay/ZnO/Ag-1 and clay/ZnO/Ag-2 recorded a 99.91% and 99.94% respective efficiency in the removal of *S. aureus* from water. Clay/ZnO/Ag-3 had a 100% effect on the eradication of the *S. aureus* strain as shown in Figure 26.

The results obtained in the antibacteria activity of clay/CuO/Ag pellet samples and clay/ZnO/Ag pellet samples on *E. coli* and *S. aureus* strains after 48 h of incubation in sterile water are illustrated in the histogram of Figure 27. All the various pellet samples including: clay/CuO/Ag-1, clay/CuO/Ag-2, clay/CuO/Ag-3, clay/ZnO/Ag-1 clay/ZnO/Ag-2 and clay/ZnO/Ag-3 showed respectively 99.94%, 100%, 100%, 99.71%, 99.85% and 100% against E. coli after 48 h of antibacteria activity. Again the activities of clay/CuO/Ag-1, clay/CuO/Ag-2, clay/CuO/Ag-3, clay/ZnO/Ag-1 clay/ZnO/Ag-2 and clay/ZnO/Ag-3 against *S. aureus* was 100%, 100%, 100%, 99.91%, 99.94% and 100%. Meanwhile, comparing with the results obtained in the antibacteria activity of clay/CuO/Ag and clay/ZnO/Ag against the same bacteria species and strains after 24 h of incubation in sterile water, clay/ZnO/Ag-1 and clay/ZnO/Ag-2 had a negligible effect. Therefore, the high results obtained by clay/ZnO/Ag-1 and clay/ZnO/Ag-2 after 48 h of incubation confirm that the antibacterial activity of the pellets was dependent on time. The time-dependent nature of the antibacterial activity of the clay–nanocomposite pellets agrees with the antibacteria activity of CuO, ZnO, CuO/Ag and ZnO/Ag also investigated in other studies [19,20,33].

The application of clay–nanocomposite pellets for water purification revealed that the material composition of the nanocomposites used, the environment of the bacteria species, and the time allowed for antibacterial activity are critical factors to consider in the design of systems for bacterial decontamination of water sources. While clay/CuO/Ag pellet samples performed better than clay/ZnO/Ag pellet samples among the different conditions of testing, they performed differently in nutrient rich environment from environment void of bacteria growth nutrient. The relative performance of the samples were also dependent on dose of nanocomposite within pellet and also the time required for antibacterial activity.

## 5. Conclusions

We have shown that a relatively cost effective clay–nanocomposite pellet can be used as an efficient medium for bacteria decontamination of water sources operational at the point of water delivery. However, the potency of antibacterial activity is dependent on the elemental composition of the nanocomposites used as fillers for the clay materials for the preparation of clay–nanocomposite pellets and the environment of the bacteria.

Therefore, the results show clay to be a very effective membrane support material for metaloxide/Ag NPs bactericides via a facile and low-cost preparatory method with high potential, and easy scalability, for efficient water purification.

## Figures and Tables

**Figure 1 materials-13-03793-f001:**
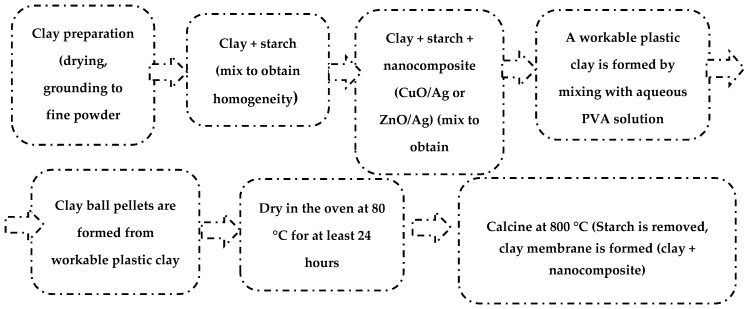
Annotated diagram for the synthesis of clay-nanocomposites.

**Figure 2 materials-13-03793-f002:**
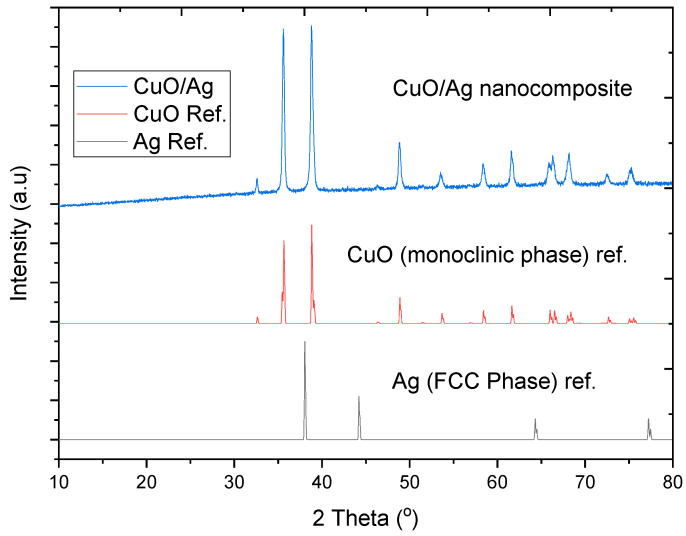
X-ray diffraction (XRD) spectrum of CuO/Ag correlated with the reference data of CuO monoclinic phase, adapted from Ref. [20] and the face-centered cubic (FCC) phase of Ag, adapted from reference [23].

**Figure 3 materials-13-03793-f003:**
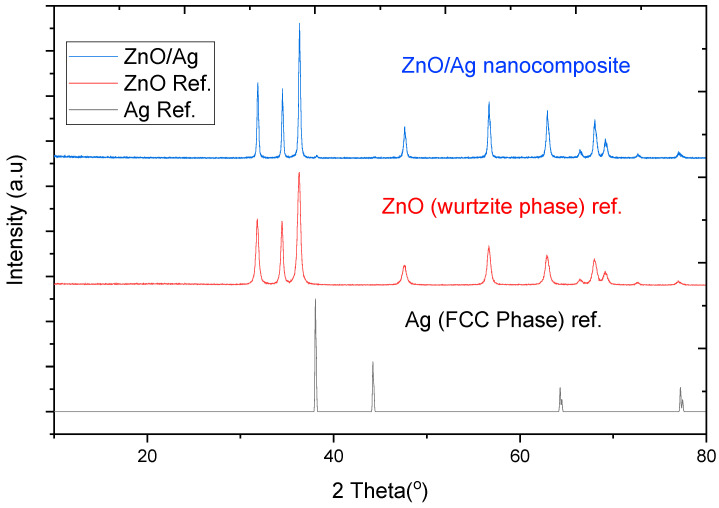
XRD pattern of ZnO/Ag nanocomposite with correlation with the wurtzite phase of ZnO adapted from Ref. [23] and the face-centered cubic (FCC) phase of Ag, adapted from reference [20].

**Figure 4 materials-13-03793-f004:**
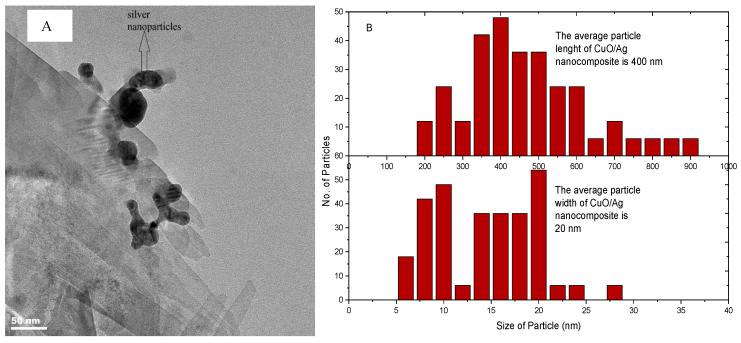
(**A**) Transmission electron microscope (TEM) micrograph of CuO/Ag and (**B**) particle size distribution of CuO/Ag nanocomposite, adapted from reference [19].

**Figure 5 materials-13-03793-f005:**
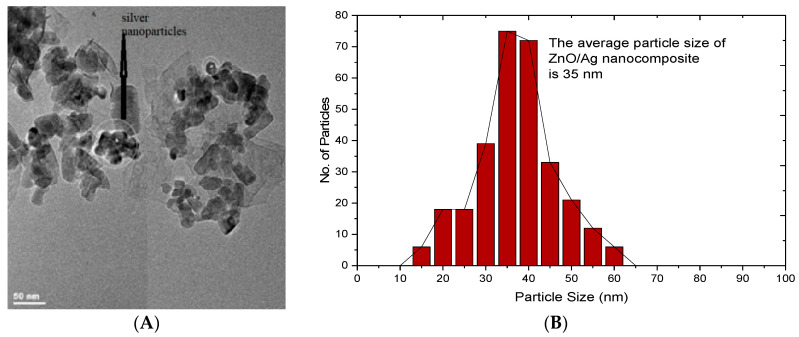
(**A**) TEM micrograph of ZnO/Ag (**B**) Particle size distribution of ZnO/Ag nanocomposite adapted from reference [19].

**Figure 6 materials-13-03793-f006:**
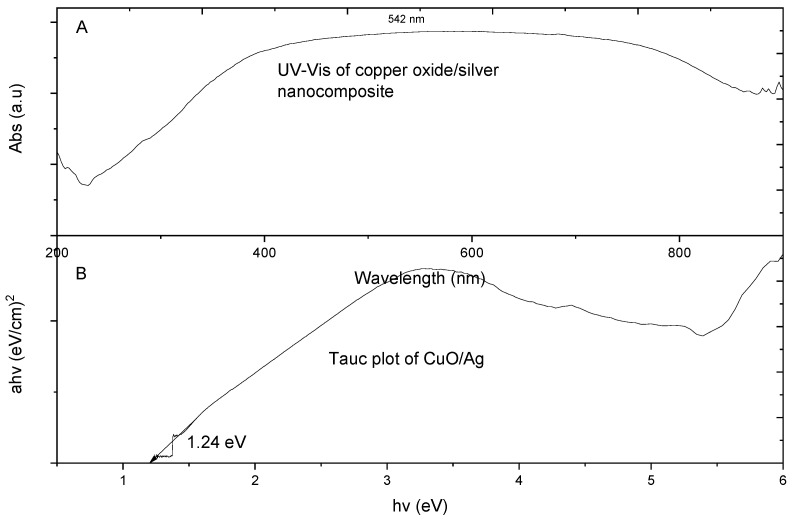
(**A**) Ultraviolet–visible (UV–Vis) spectrophotometric absorption spectra of CuO/Ag and (**B**) Tauc plot of CuO/Ag, adapted from reference [19].

**Figure 7 materials-13-03793-f007:**
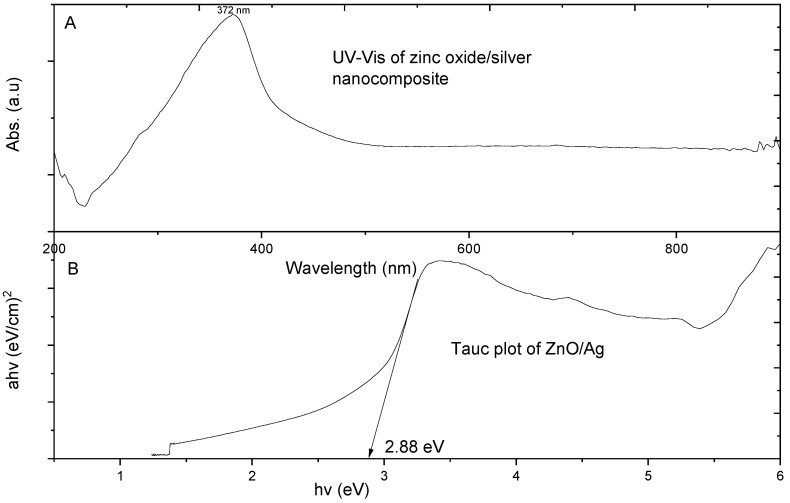
UV–Vis spectra of ZnO/Ag nanocomposite with its equivalent plot showing its energy band gap, adapted from reference [19]. (**A**)-Ultraviolet–visible (UV–Vis) spectrophotometric absorption spectra of ZnO/Ag and (**B**)-Tauc plot of ZnO/Ag.

**Figure 8 materials-13-03793-f008:**
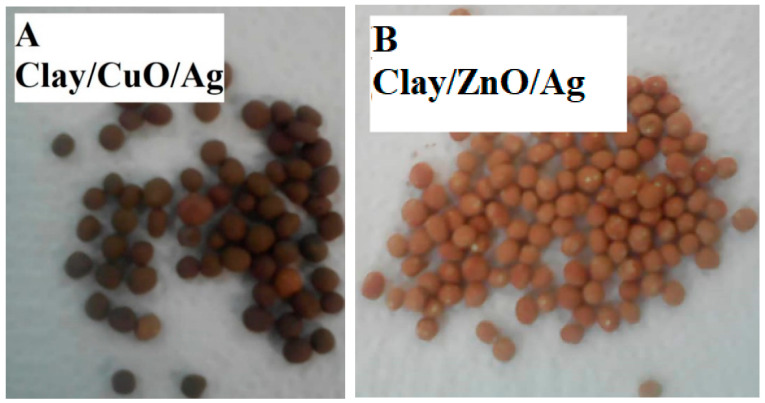
Synthesised clay–nanocomposite pellets of (**A**) CuO/Ag nanocomposite (**B**) ZnO/Ag nanocomposite.

**Figure 9 materials-13-03793-f009:**
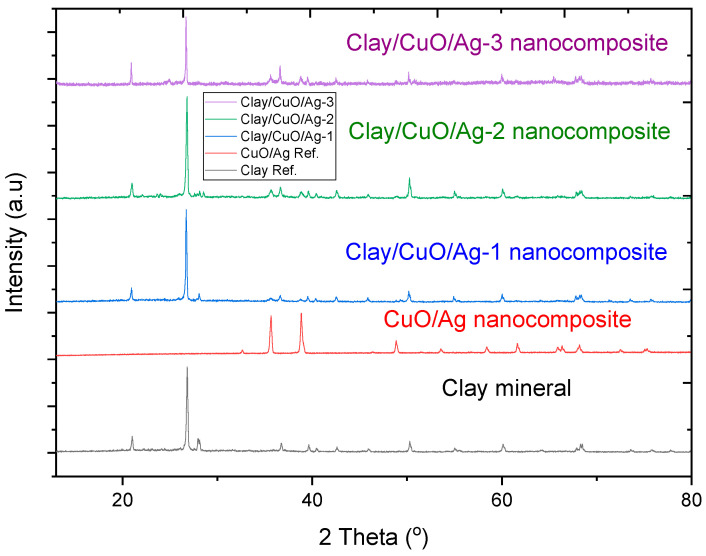
XRD spectra pattern of clay mineral, CuO/Ag, clay/CuO/Ag. Clay/CuO/Ag-1, clay/CuO/Ag-2, clay/CuO/Ag-3 contain 2.5%, 5% and 10% of CuO/Ag, respectively.

**Figure 10 materials-13-03793-f010:**
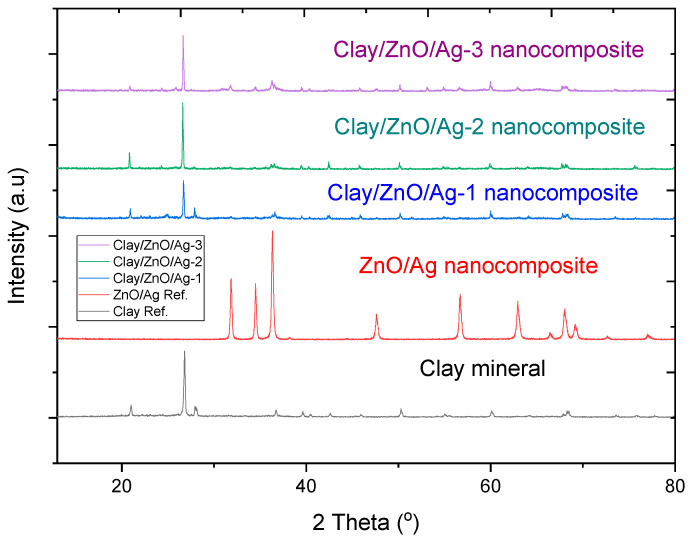
XRD spectra pattern of clay mineral, ZnO/Ag, clay/ZnO/Ag. Clay/ZnO/Ag-1, clay/ZnO/Ag-2, clay/ZnO/Ag-3 have 2.5%, 5% and 10% of ZnO/Ag, respectively.

**Figure 11 materials-13-03793-f011:**
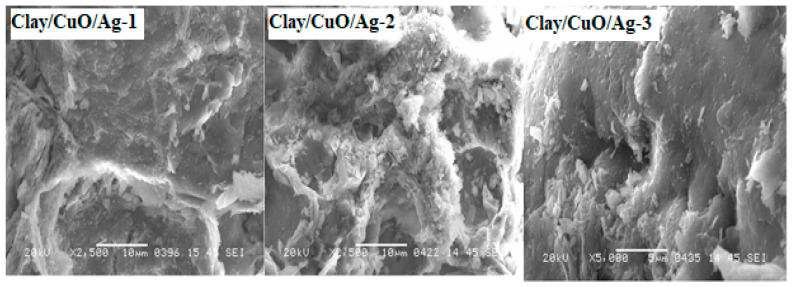
Scanning electron microscope (SEM) micrograph of clay/CuO/Ag-1, clay/CuO/Ag-2 and clay/CuO/Ag-3.

**Figure 12 materials-13-03793-f012:**
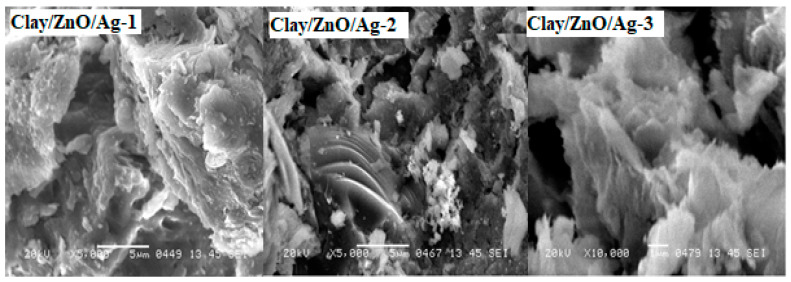
SEM micrograph of clay/ZnO/Ag-1, clay/ZnO/Ag-2 and clay/ZnO/Ag-3.

**Figure 13 materials-13-03793-f013:**
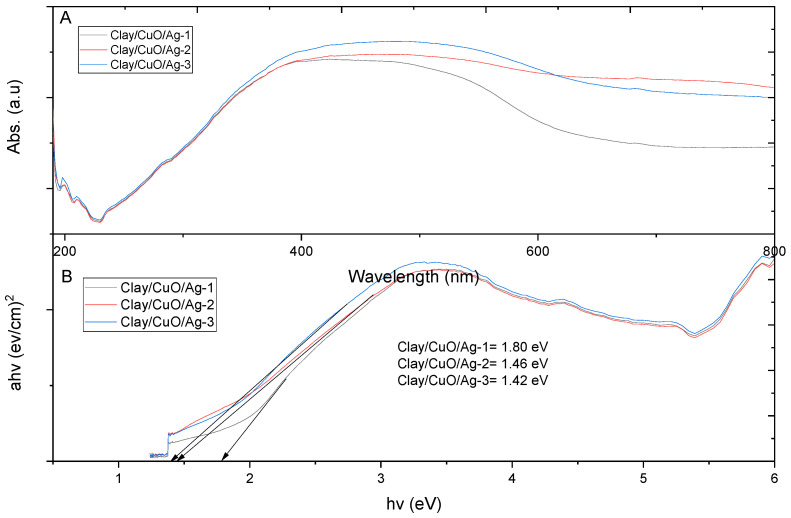
(**A**) UV–Vis spectrophotometric absorption spectra of clay/CuO/Ag-1, clay/CuO/Ag-2 and clay/CuO/Ag-3 and (**B**) their corresponding band gap plots.

**Figure 14 materials-13-03793-f014:**
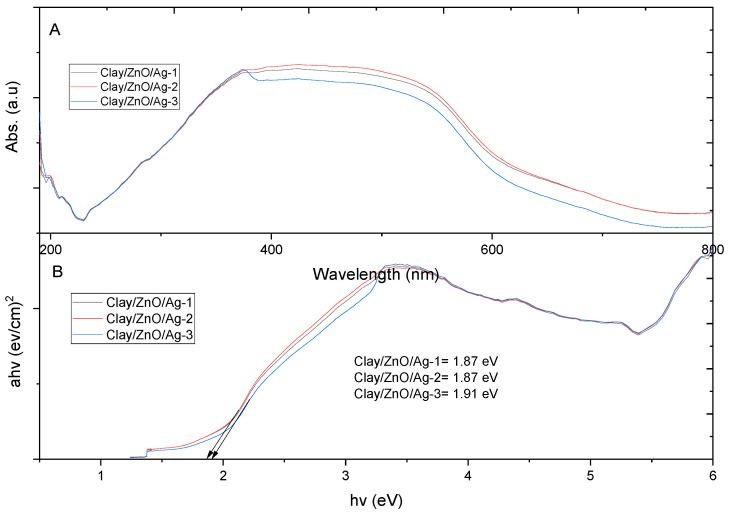
(**A**) UV–Vis spectrophotometric absorption spectra of clay/ZnO/Ag-1, clay/ZnO/Ag-2 and clay/ZnO/Ag-3 and (**B**) their respective band gap plots.

**Figure 15 materials-13-03793-f015:**
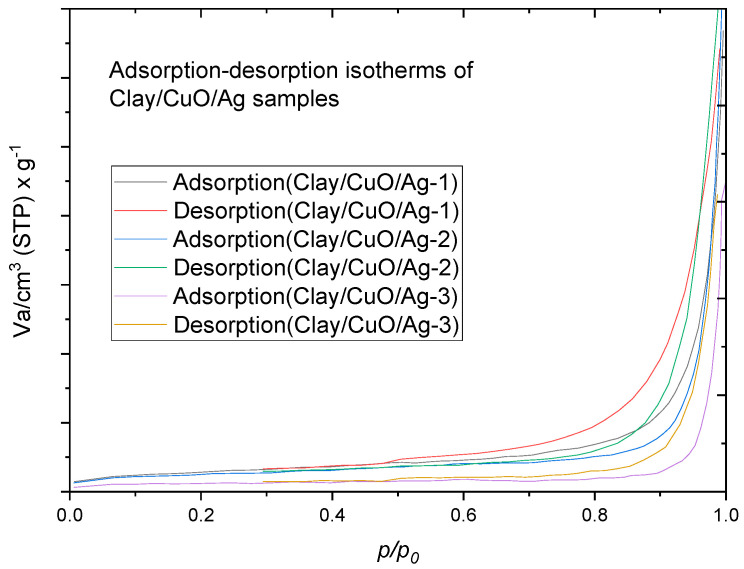
Adsorption-desorption isotherms of nitrogen gas at 77 K of “as-received” pellets of clay/CuO/Ag samples.

**Figure 16 materials-13-03793-f016:**
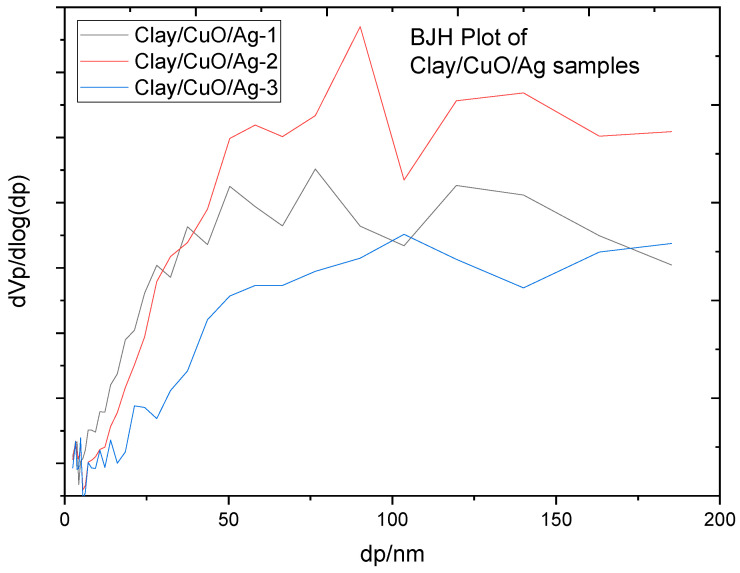
Pore size distribution of clay/CuO/Ag samples.

**Figure 17 materials-13-03793-f017:**
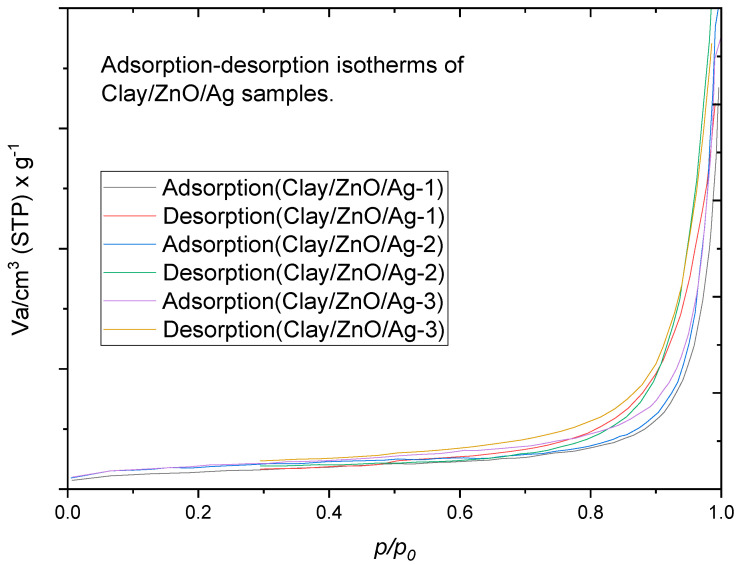
Adsorption-desorption isotherms of nitrogen gas at 77 K of “as-received” pellets of clay/ZnO/Ag samples.

**Figure 18 materials-13-03793-f018:**
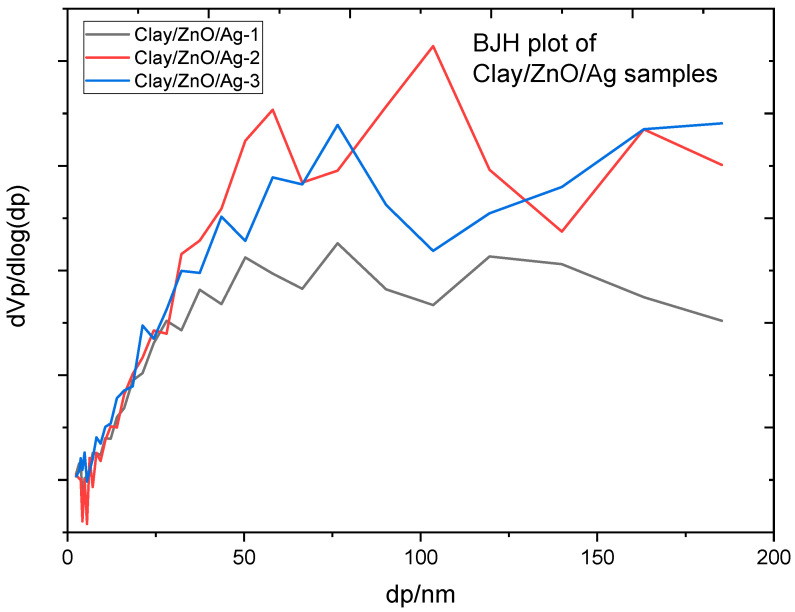
Pore size distribution of clay/CuO/Ag samples.

**Figure 19 materials-13-03793-f019:**
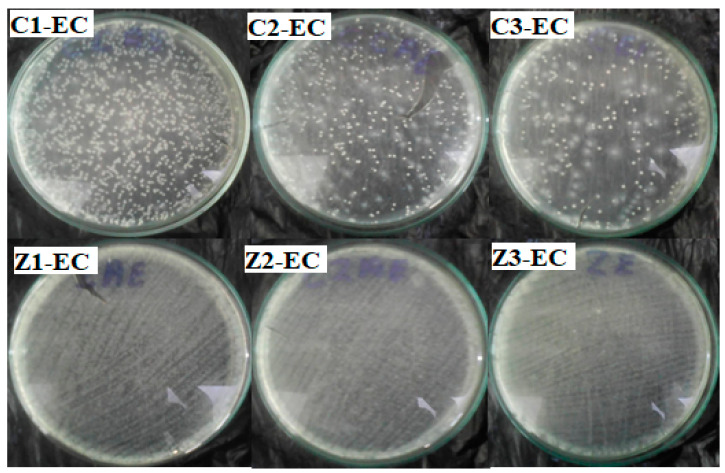
Antibacterial activity of clay/CuO/Ag and clay/ZnO/Ag against *E. coli.* C1-EC, C2-EC, C3-EC correspondingly relates to clay/CuO/Ag-1, clay/CuO/Ag-2 and clay/CuO/Ag-3 against *E. coli.* Z1-EC, Z2-EC, Z3-EC correspondingly relates to clay/ZnO/Ag-1, clay/ZnO/Ag-2 and clay/ZnO/Ag-3 against *E. coli.*

**Figure 20 materials-13-03793-f020:**
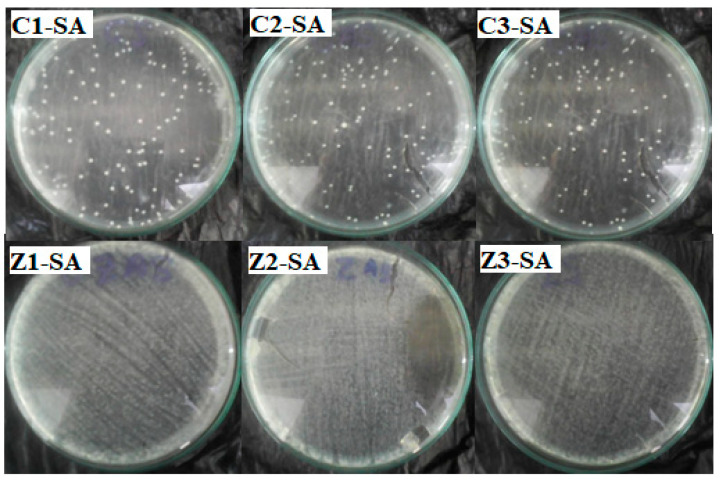
Antibacteria activity of clay/CuO/Ag and clay/ZnO/Ag against *S. aureus*. C1-EC, C2-EC, C3-EC correspondingly relates to clay/CuO/Ag-1, clay/CuO/Ag-2 and clay/CuO/Ag-3 against *S. aureus*. Z1-EC, Z2-EC, Z3-EC correspondingly relates to clay/ZnO/Ag-1, clay/ZnO/Ag-2 and clay/ZnO/Ag-3 against *S. aureus*.

**Figure 21 materials-13-03793-f021:**
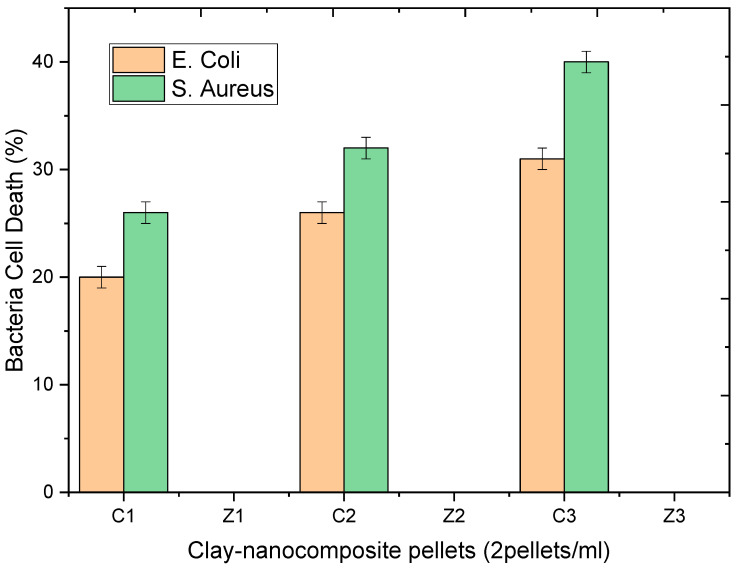
Histogram of the antibacteria activity of clay–nanocomposites pellets against *E. coli* and *S. aureus* in broth media after 24 h of incubation. C1, C2, C3, Z1, Z2 and Z3 respectively relates to: clay/CuO/Ag-1, clay/CuO/Ag-2, clay/CuO/Ag-3, clay/ZnO/Ag-1, clay/ZnO/Ag-2 and clay/ZnO/Ag-3 (Z1, Z2 and Z3 are empty because there was no any recorded inhibition of the bacterial species in the broth media).

**Figure 22 materials-13-03793-f022:**
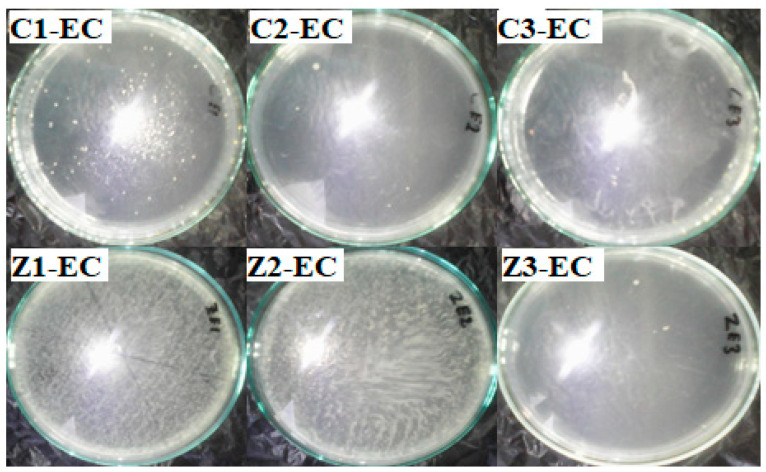
Antibacteria activity of clay/CuO/Ag and clay/ZnO/Ag against E. coli. C1-EC, C2-EC, C3-EC correspondingly relates to clay/CuO/Ag-1, clay/CuO/Ag-2 and clay/CuO/Ag-3 against E. coli. Z1-EC, Z2-EC, Z3-EC correspondingly relates to clay/ZnO/Ag-1, clay/ZnO/Ag-2 and clay/ZnO/Ag-3 against *E. coli*.

**Figure 23 materials-13-03793-f023:**
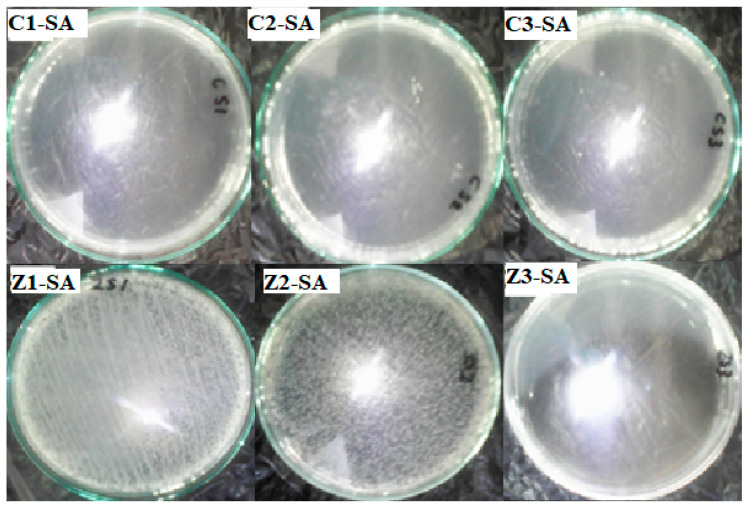
Antibacteria activity of clay/CuO/Ag and clay/ZnO/Ag against *S. aureus* in water after 24 h of incubation. C1-EC, C2-EC, C3-EC correspondingly relates to clay/CuO/Ag-1, clay/CuO/Ag-2 and clay/CuO/Ag-3 against *S. aureus*. Z1-EC, Z2-EC, Z3-EC correspondingly relates to clay/ZnO/Ag-1, clay/ZnO/Ag-2 and clay/ZnO/Ag-3 against *S. aureus*.

**Figure 24 materials-13-03793-f024:**
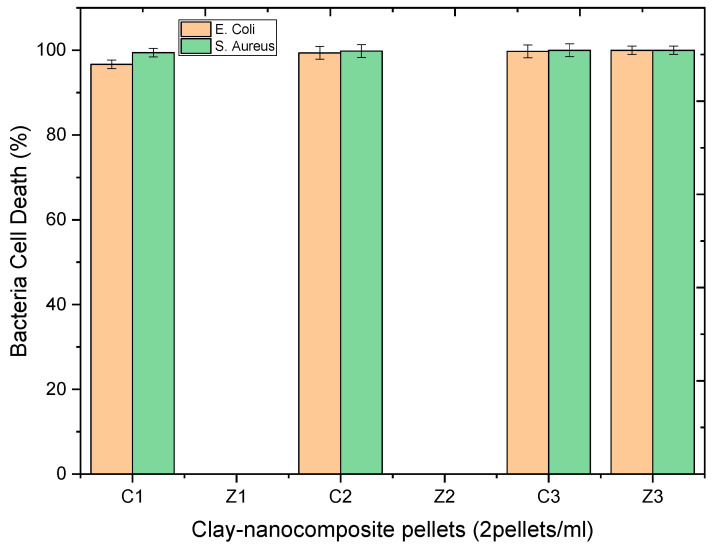
Histogram of the antibacteria activity of clay–nanocomposites pellets against *E. coli* and *S. aureus* in sterile aqueous media after 24 h of incubation. C1, C2, C3, Z1, Z2 and Z3 respectively relates to: clay/CuO/Ag-1, clay/CuO/Ag-2, clay/CuO/Ag-3, clay/ZnO/Ag-1, clay/ZnO/Ag-2 and clay/ZnO/Ag-3. (Z1 and Z2 are empty because there was no any recorded inhibition of the bacterial species in the aqueous media).

**Figure 25 materials-13-03793-f025:**
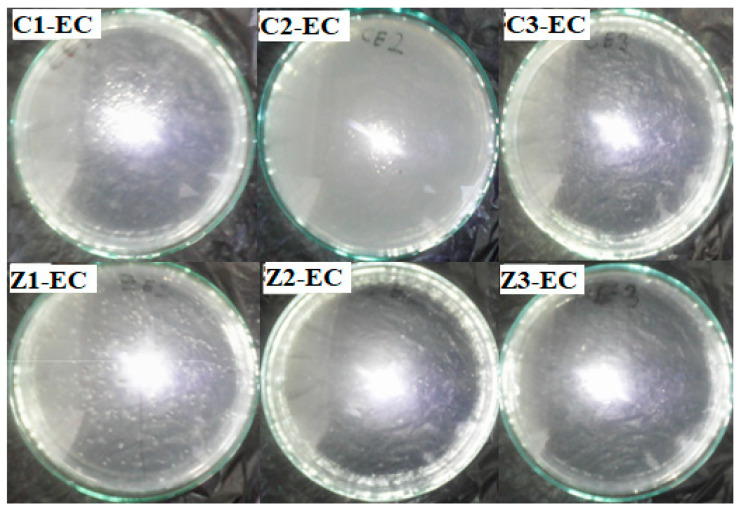
Antibacteria activity of clay/CuO/Ag and clay/ZnO/Ag against *E. coli* in water after 48 h of incubation. C1-EC, C2-EC, C3-EC correspondingly relates to clay/CuO/Ag-1, clay/CuO/Ag-2 and clay/CuO/Ag-3 E. coli. Z1-EC, Z2-EC, Z3-EC correspondingly relates to clay/ZnO/Ag-1, clay/ZnO/Ag-2 and clay/ZnO/Ag-3 against *E. coli*.

**Figure 26 materials-13-03793-f026:**
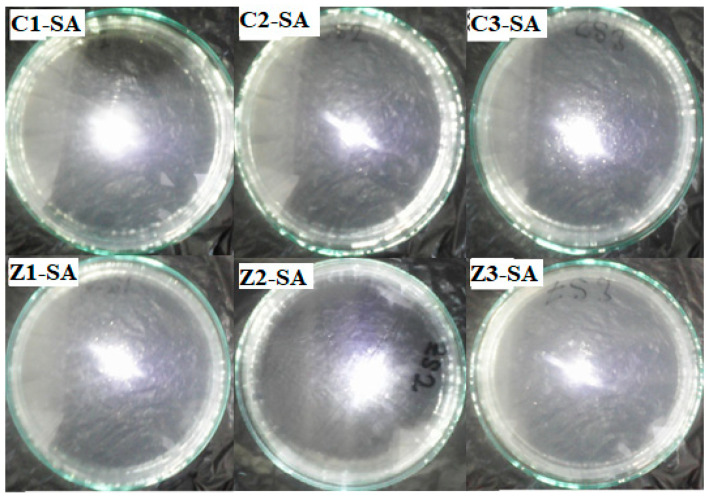
Antibacteria activity of clay/CuO/Ag and clay/ZnO/Ag against *S. aureus* in water after 48 h of incubation. C1-EC, C2-EC, C3-EC correspondingly relates to clay/CuO/Ag-1, clay/CuO/Ag-2 and clay/CuO/Ag-3 against *S. aureus*. Z1-EC, Z2-EC, Z3-EC correspondingly relates to clay/ZnO/Ag-1, clay/ZnO/Ag-2 and clay/ZnO/Ag-3 against *S. aureus*.

**Figure 27 materials-13-03793-f027:**
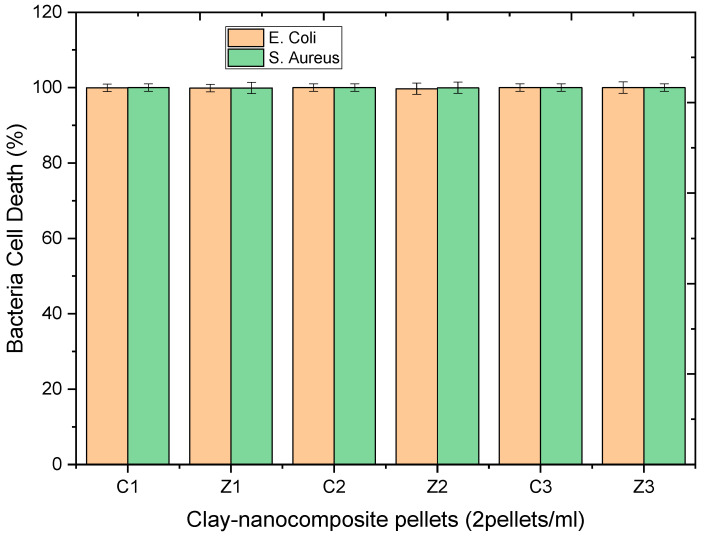
Histogram of the antibacteria activity of clay–nanocomposites pellets against *E. coli* and *S. aureus* in sterile aqueous media after 48 h of incubation. C1, C2, C3, Z1, Z2 and Z3, respectively relates to: clay/CuO/Ag-1, clay/CuO/Ag-2, clay/CuO/Ag-3, clay/ZnO/Ag-1, clay/ZnO/Ag-2 and clay/ZnO/Ag-3.

**Table 1 materials-13-03793-t001:** Weight of materials used for preparing clay membranes A, B and C.

Clay Nanocomposite (Clay Pellet)	Clay Mineral	Nanocomposite (CuO/Ag or ZnO/Ag)	Starch
	Weight (g)	Weight (g)	Weight (g)
A	20	0.7	2.8
B	20	1.4	2.8
C	20	2.9	2.8

**Table 2 materials-13-03793-t002:** X-ray fluorescence (XRF) analysis of the components of Copper Oxide/Silver and zinc oxide/silver (ZnO/Ag) nanocomposites (in wt.%), adapted from reference [19].

Component.	CuO/Ag	ZnO/Ag
Al_2_O_3_	0.0849	0.105
SiO_2_	0.147	0.257
P_2_O_5_	0.0146	0.0203
K_2_O	0.0429	0.0473
CaO	0.0214	0.0216
CuO	99.2	
ZnO		98.8
Ag_2_O	0.518	0.547

**Table 3 materials-13-03793-t003:** X-ray fluorescence (XRF) analysis of the components of clay/CuO/Ag samples: clay/CuO/Ag-1, clay/CuO/Ag-2 and clay/CuO/Ag-3 nanocomposites (in wt.%).

Component	Clay/CuO/Ag-1	Clay/CuO/Ag-2	Clay/CuO/Ag-3
MgO	0.733	0.699	0.667
Al_2_O_3_	19.9	19.5	18.2
SiO_2_	65.4	64.2	62.4
P_2_O_5_	0.234	0.229	0.217
Cl	0.0093	0.0102	0.0107
K_2_O	0.988	0.939	0.918
CaO	0.438	0.424	0.437
TiO_2_	0.957	0.93	0.894
MnO	0.0524	0.0505	0.0509
Fe_2_O_3_	6.4	5.99	5.76
CuO	3.46	5.25	8.97
Ag_2_O	0.0102	0.0114	0.0185
Na_2_O	1.45	1.72	1.53

**Table 4 materials-13-03793-t004:** XRF analysis of the components of clay/ZnO/Ag samples: clay/ZnO/Ag-1, clay/ZnO/Ag-2 and clay/ZnO/Ag-3 nanocomposites (in wt.%).

Component	Clay/ZnO/Ag-1	Clay/ZnO/Ag-1	Clay/ZnO/Ag-1
MgO	0.732	0.624	0.653
Al_2_O_3_	18.9	18.2	16.4
SiO2	62.2	62.9	51.8
P_2_O_5_	0.216	0.217	0.179
K2O	0.978	0.919	0.797
CaO	0.398	0.402	0.316
TiO2	0.915	0.886	0.786
MnO	0.0658	0.0638	0.0596
Fe2O3	6.49	6.26	5.79
ZnO	4.66	5.15	12.7
Ag2O	0.0148	0.0198	0.061
Na2O	4.42	4.33	10.3

**Table 5 materials-13-03793-t005:** Brunauer–Emmett–Teller (BET) surface area, pore volume and peak pore size of clay–nanocomposite pellets.

Clay–Nanocomposite Pellet	Surface Area (S_BET_)	Pore Volume (Vm)	Peak Pore Size (dp)
	(m^2^/g)	(cm^3^/g)	(nm)
Clay/CuO/Ag			
Clay/CuO/Ag-1	2.09	0.4807	76.49
Clay/CuO/Ag-2	3.05	0.7	90.22
Clay/CuO/Ag-3	4.25	0.9773	103.54
Clay/ZnO/Ag			
Clay/ZnO/Ag-1	5.08	1.1667	76.49
Clay/ZnO/Ag-2	6.39	1.4678	103.54
Clay/ZnO/Ag-3	6.46	1.4845	185.28

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
