# Peer review of "Development and Characterization of Clay–Nanocomposites for Water Purification"

_materials, 2020, doi:10.3390/ma13173793_

Round 1
Reviewer 1 Report
Dear author,
The paper Development and Characterization of Clay-Nanocomposites for Water Purification deals with interesting and important theme especially in poorer economy countries. The property of clay to catch some impurities from environment is known, but the combination of clay with metal oxides is interesting idea. The paper is good quality with good experiment explanation.
Just several comments for improving:
- The paper is interesting but too long. It could be divided into two papers – one dealing with Clay-CuO/Ag and second with Clay-ZnO/Ag materials. The final decision is up to the authors of the paper.
- In Part 2 Materials and Methods – 1.2.1 What does the abbreviation LB mean?
- In Part 2 Materials and Methods - Please check the chapter numbering. There is probably a mistake in this chapter.
- In Part 2 Materials and Methods – I miss the description of XRD, XRF, UV-VIS, SEM measurement conditions.
- How was obtained band gap from UV-VIS measurement?
- Fig. 16 and 17 – try to use brighter curve colors or thicker lines.
- Fig. 22, 25 – Why information related to Z1, Z2, Z3 are missing in the graphs?
- Fig. 28 – Do you really obtain the same results for all materials?
Best regards
Reviewer
Author Response
Reviewer 1
Comments and Suggestions for Authors
Dear author,
The paper Development and Characterization of Clay-Nanocomposites for Water Purification deals with interesting and important theme especially in poorer economy countries. The property of clay to catch some impurities from environment is known, but the combination of clay with metal oxides is interesting idea. The paper is good quality with good experiment explanation.
Just several comments for improving:
- The paper is interesting but too long. It could be divided into two papers – one dealing with Clay-CuO/Ag and second with Clay-ZnO/Ag materials. The final decision is up to the authors of the paper.
Answer:
Authors are most grateful for this comment. But dividing the manuscript into two papers becomes a very difficult task since the paper is comparing the antibacterial activity of two nanoparticles; Clay-CuO/Ag and Clay-ZnO/Ag materials.
- In Part 2 Materials and Methods – 1.2.1 What does the abbreviation LB mean?
Answer:
Revised text: Luria-Bertani (LB) broth
Authors’ response: We apologize for the oversight.
- In Part 2 Materials and Methods - Please check the chapter numbering. There is probably a mistake in this chapter.
Answer: We apologize for the oversight and it has been rectified in the manuscript.
Revised text: 2. Materials and Methods
2.1 Synthesis of Nanocomposites
- In Part 2 Materials and Methods – I miss the description of XRD, XRF, UV-VIS, SEM measurement conditions.
We thank the reviewer for the comments. All the measurement conditions have been included in the manuscript.
- How was obtained band gap from UV-VIS measurement?
Answer:
The bandgaps for the samples were estimated by a Tauc Plot using the UV-Vis absorbance data obtained by measuring suspensions/colloids of the samples in solution with UV-Vis Instrument via a transmission mode. The quantity hν was plotted against quantity (αhν)r
Bandgaps of the respective nanocomposites were obtained from the Tauc plot by extrapolating the line on the curve to the corresponding x-axis (hν).
- 16 and 17 – try to use brighter curve colors or thicker lines.
Answer: This has been done.
- 22, 25 – Why information related to Z1, Z2, Z3 are missing in the graphs?
Answer:
These were empty because there was no any recorded inhibition of the bacterial species in the broth media (Z1, Z2 & Z3).
- 28 – Do you really obtain the same results for all materials?
Answer
No, this is because, the materials percent compositions exhibited different activity on the bacterial species.
Reviewer 2 Report
The following questions should be adequately answered:
The choice and use of CuO/Ag and ZnO/Ag must be explained. If there are previous works of other authors that justify the use of these metal compositions, these works must be reviewed and discussed in the manuscript. The authors indicate that they have used the method proposed in reference 23; the methodology used in the present work must be included and described step by step. How do the authors justify the chosen metal relationships and the composition presented in Table 1?
The information that allows the determination of the crystallite size through the Scherrer equation must be added.
What are the characteristics of the clays?
Figures 5 and 6: to which metal do the particle distributions correspond?
Figures 7 and 8: to what does “tauc plot” correspond?
Point 3.2.5 Brunauer ... must be rewritten, because it is not correct. The adsorption isotherms are of type II. The textural results included in Table 4 are also not correct. The specific surfaces must appear without decimals and, if the adsorption isotherms are type II, the total pore volume values should be eliminated.
Author Response
August 02, 2020
Dear Editor-in-Chief,
We are very grateful the comments received on our paper.
It is our belief that we have improved upon the manuscript considerably and is now suitable for publication in your journal.
We also appreciate the efforts of reviewer 1 & 2, for their deep insights which have resulted in improving our manuscript.
Thank you.
Onwona-Agyeman, PhD
On behalf of all authors.
Reviewer 3 Report
The paper represents a significant study of the new clay-nanocomposites material for water treatment. The low cost and high-efficiency properties make it becoming an attractive method in the water treatment industry. I think, nonetheless, that the manuscript could be improved if the authors could address the comments and recommendations I listed below.
- The novelty of this research should be highlighted in the Abstract.
- In section 2.2.2. Your bentonite clay was obtained locally, and followed by some treatment. Did you do the chemical composition analysis of your bentonite clay before and after your treatment? Some chemicals may affect water treatment.
- In section 3.1.1. Since your CuO and ZnO were about 99% in their system. I agree with you that XRD can not detect Ag. In my opinion, the XRD measurement in this section is not necessary since XRD can not tell anything. But, in section 3.2.1, XRD measurement is important since it indications the success of mixture.
- Line 216. Not sure how you concluded the average length from Figure 3.
- A typo in Figure 5 A.
- Line 250-252: Give a citation.
- Figure 22, 23. No data on Z1, Z2, and Z3. You should insert all your labeled data.
- Figure 22, 25, and 28. What's the difference between them. You should make is more clear to the reader. Maybe you can add some words in the figure caption to help people distinguish them. Figure 28 should be the date after 48 hours, but you did not change the figure caption.
- Based on your data, the cell death rate can reach to around 100% after 48hrs incubation. It looks pretty nice. However, as you mentioned at the last, "antibacterial activity was dependent on the elemental composition of the nanocomposites". Have you done some related research?
Author Response
Reviewer 2
Comments and Suggestions for Authors
The paper represents a significant study of the new clay-nanocomposites material for water treatment. The low cost and high-efficiency properties make it becoming an attractive method in the water treatment industry. I think, nonetheless, that the manuscript could be improved if the authors could address the comments and recommendations I listed below.
- The novelty of this research should be highlighted in the Abstract.
We thank the reviewer for this comment.
Answer:
This work reports on the use of facile and cost effective preparation of very efficient nanocomposites of Clay/ZnO/Ag and Clay/CuO/Ag by the use of naturally occurring bentonite and wet chemical synthesis.
- In section 2.2.2. Your bentonite clay was obtained locally, and followed by some treatment. Did you do the chemical composition analysis of your bentonite clay before and after your treatment? Some chemicals may affect water treatment.
This is actually right. We are grateful to the reviewer for the comment.
Answer: The chemical composition of the clay was only carried out after treatment in order to ascertain the effect on the water treatment.
- In section 3.1.1. Since your CuO and ZnO were about 99% in their system. I agree with you that XRD cannot detect Ag. In my opinion, the XRD measurement in this section is not necessary since XRD cannot tell anything. But, in section 3.2.1, XRD measurement is important since it indications the success of mixture.
We are grateful for this comment which is true.
Answer: XRD was performed to evaluate the crystalline microstructure of the samples
- Line 216. Not sure how you concluded the average length from Figure 3.
Answer:
The ‘average length’ was estimated from the Scherer’s equation which is normally used to determine the crystallite size or length of materials using XRD measurements. A reference paper has been added to this effect.
- A typo in Figure 5 A.
We thank the reviewer for spotting the typo.
Answer: This has been corrected.
- Line 250-252: Give a citation.
Answer: This has been done.
- Figure 22, 23. No data on Z1, Z2, and Z3. You should insert all your labeled data.
Answer: These were missing because there was no any recorded inhibition of the bacterial species in the broth media (Z1, Z2 & Z3).
- Figure 22, 25, and 28. What's the difference between them. You should make is more clear to the reader. Maybe you can add some words in the figure caption to help people distinguish them. Figure 28 should be the date after 48 hours, but you did not change the figure caption.
Answer: This has been corrected
Based on your data, the cell death rate can reach to around 100% after 48hrs incubation. It looks pretty nice. However, as you mentioned at the last, "antibacterial activity was dependent on the elemental composition of the nanocomposites". Have you done some related research?
Yes, please refer to the following:
- Asamoah RB, Mensah B, Annan E, Nbelayim P, Appalanya V, Onwona-Agyeman B, Yaya A. A Comparative study of antibacterial activity of CuO/Ag and ZnO/Ag Nanocomposites. Advances in Material Science and Engineering, (2020).
- Asamoah RB, Yaya A, Mensah B, Nbelayim P, Apalangya V, Yaw DB, Damoah LNW, Agyei-Tuffour B, Dodoo-Arhin D, Annan E . Synthesis and characterization of zinc and copper oxide nanoparticles and their antibacterial activity. Results in Materials, (2020).
Reviewer 4 Report
This work investigated the antibacteria activity of local clay materials with metal-oxide based nanocomposites. This contributes to the development of clay-nanocomposites for water purification. However, in the result and discussion section, the characteristics of the samples are listed, but the influence of these characteristics on the antibacteria activity is not analyzed. Besides, some problems also need to be addressed.
- Please make your abstract with more specific and quantitative results while it suits broader audiences.
- The originality of the paper needs to be further clarified. It is of importance to have sufficient results to justify the novelty of this paper. It is necessary to supplement the existing articles on clay membranes incorporated by antibacteria nanoparticles to clarify the novelty of this work.
- Why use Cu0/Ag and Zn0/Ag as additive nanoparticles? What are the advantages of these metal oxides compared with other metal oxides?
- The contribution of this study should be clarified in last paragraph in the Introduction
- Line 176-177: The (XRF) further confirms the purity of the samples which is critical to determine the true antibacterial effect of the nanocomposite antibacterial agents.
What is critical to determine the true antibacterial effect of the nanocomposite antibacterial agents?
- Result and discussion section: The characteristics of the samples are listed, but the influence of these characteristics on the antibacteria activity is not analyzed
- It is suggested to integrate the same characterization in 3.1 and 3.2. The characteristics changes before and after adding oxides should be compared. And the effect of these changes on the antibacteria activity should be analyzed.
- The colclusion section is too long. It is suggested to list the important conclusions briefly
- Conclusion: The amount of toxic metal ions present in solution increased with increasing concentration of respective nanocomposite within clay hence higher antibacteria activity.
As mentioned in the conclusion, why increase the concentration of nanocomposites? This will increase the concentration of toxic metal ions in the solution.
- The conclusion should focus on the effect of metal oxides addition on antibacterial activity of clay.
Author Response
July 27, 2020
Dear Editor-in-Chief,
We are very grateful for the comments and results on the similarity index check on our manuscript. We noticed that the 16 % recorded on the index check was as a result of our previously published works and similar wording/names which are commonly used scientific terms. To this effect, we have managed to reconstruct our sentences in the manuscript in order to lower the index considerably.
It is our belief that we have improved upon the manuscript considerably and is now suitable for publication in your journal.
We also appreciate the efforts of reviewers for their deep insights which have resulted in improving our manuscript.
Please find below our responses to your comments raised earlier in Orange/yellow.
Thank you.
- Onwona-Agyeman, PhD
On behalf of all authors.
Round 2
Reviewer 3 Report
The authors have satisfactorily responded to all my questions.
Author Response
We are most grateful to the reviewer for his time, thoughts and insight, for all the inputs and suggestions on our manuscript. Below are our responses to his comments.
Thank you.
Reviewer 2
Comments and Suggestions for Authors
In the revised manuscript, parts of the problems have been addressed. However, the important problems have not been revised.
- Please make your abstract with more quantitative results to show the antibacteria activity of Clay-Nanocomposite.
We are most grateful for your thoughts and insightful comments.
Answer: The abstract has been completely modified to reflect the reviewer’s comments
- The originality of the paper was not clarified in the revised manuscript. It is of importance to have sufficient results to justify the novelty of this paper. It is necessary to supplement the existing articles on clay membranes incorporated by antibacteria nanoparticles to clarify the novelty of this work.
Thank you for the comments. We have tried our best to answer this question.
Answer: This suggestion has been implemented in both the abstract and introduction. The conclusion has also been re-written to capture this comment
- Why use Cu0/Ag and Zn0/Ag as additive nanoparticles? The advantages of these metal oxides compared with other metal oxides should be clarified in the Introduction.
We thank the reviewer for this comment.
Answer: This has been done. Please refer to the introduction where the below section has been added in the text.
Meanwhile, copper oxide/silver (CuO/Ag) and zinc oxide/silver (ZnO/Ag) nanocomposites have independently shown broad-spectrum and a highly efficient antibacterial activities relative to their respective separate metal oxide and metal nanoparticle counterparts [19-20].
Reviewer 4 Report
In the revised manuscript, parts of the problems have been addressed. However, the important problems have not been revised.
- Please make your abstract with more quantitative results to show the antibacteria activity of Clay-Nanocomposite.
- The originality of the paper was not clarified in the revised manuscript. It is of importance to have sufficient results to justify the novelty of this paper. It is necessary to supplement the existing articles on clay membranes incorporated by antibacteria nanoparticles to clarify the novelty of this work.
- Why use Cu0/Ag and Zn0/Ag as additive nanoparticles? The advantages of these metal oxides compared with other metal oxides should be clarifed in the Introduction.
Author Response

(The authors gave the same response as above.)
